# Structural and biophysical analysis of a *Haemophilus influenzae* tripartite ATP-independent periplasmic (TRAP) transporter

Michael J Currie[1][†], James S Davies[1,2][†], Mariafrancesca Scalise[3], Ashutosh Gulati[2], Joshua D Wright[1], Michael C Newton-Vesty[1], Gayan S Abeysekera[1], Ramaswamy Subramanian[4], Weixiao Y Wahlgren[5], Rosmarie Friemann[6], Jane R Allison[7], Peter D Mace[8], Michael DW Griffin[9], Borries Demeler[10,11], Soichi Wakatsuki[12,13], David Drew[2], Cesare Indiveri[3], Renwick CJ Dobson[1,9]*, Rachel A North[2,14]*

[1]Biomolecular Interaction Centre, Maurice Wilkins Centre for Biodiscovery, MacDiarmid Institute for Advanced Materials and Nanotechnology, and School of Biological Sciences, University of Canterbury, Christchurch, New Zealand; [2]Department of Biochemistry and Biophysics, Stockholm University, Stockholm, Sweden; [3]Department DiBEST (Biologia, Ecologia, Scienze della Terra) Unit of Biochemistry and Molecular Biotechnology, University of Calabria, Arcavacata di Rende, Italy; [4]Biological Sciences and Biomedical Engineering, Bindley Bioscience Center, Purdue University West Lafayette, West Lafayette, United States; [5]Department of Chemistry and Molecular Biology, Biochemistry and Structural Biology, University of Gothenburg, Gothenburg, Sweden; [6]Centre for Antibiotic Resistance Research (CARe) at University of Gothenburg, Gothenburg, Sweden; [7]Biomolecular Interaction Centre, Digital Life Institute, Maurice Wilkins Centre for Molecular Biodiscovery, and School of Biological Sciences, University of Auckland, Auckland, New Zealand; [8]Biochemistry Department, School of Biomedical Sciences, University of Otago, Dunedin, New Zealand; [9]ARC Centre for Cryo-electron Microscopy of Membrane Proteins, Bio Molecular Science and Biotechnology Institute, Department of Biochemistry and Pharmacology, University of Melbourne, Melbourne, Australia; [10]Department of Chemistry and Biochemistry, University of Montana, Missoula, United States; [11]Department of Chemistry and Biochemistry, University of Lethbridge, Lethbridge, Canada; [12]Biological Sciences Division, SLAC National Accelerator Laboratory, Menlo Park, United States; [13]Department of Structural Biology, Stanford University School of Medicine, Stanford, United States; [14]School of Medical Sciences, Faculty of Medicine and Health, University of Sydney, Sydney, Australia

**\*For correspondence:**
renwick.dobson@canterbury.ac.nz (RCJD);
rachel.north@sydney.edu.au (RAN)

[†]These authors contributed equally to this work

**Abstract** Tripartite ATP-independent periplasmic (TRAP) transporters are secondary-active transporters that receive their substrates via a soluble-binding protein to move bioorganic acids across bacterial or archaeal cell membranes. Recent cryo-electron microscopy (cryo-EM) structures of TRAP transporters provide a broad framework to understand how they work, but the mechanistic details of transport are not yet defined. Here we report the cryo-EM structure of the *Haemophilus influenzae* N-acetylneuraminate TRAP transporter (*Hi*SiaQM) at 2.99 Å resolution (extending to 2.2 Å

at the core), revealing new features. The improved resolution (the previous *Hi*SiaQM structure is 4.7 Å resolution) permits accurate assignment of two Na$^+$ sites and the architecture of the substrate-binding site, consistent with mutagenic and functional data. Moreover, rather than a monomer, the *Hi*SiaQM structure is a homodimer. We observe lipids at the dimer interface, as well as a lipid trapped within the fusion that links the SiaQ and SiaM subunits. We show that the affinity ($K_D$) for the complex between the soluble *Hi*SiaP protein and *Hi*SiaQM is in the micromolar range and that a related SiaP can bind *Hi*SiaQM. This work provides key data that enhances our understanding of the 'elevator-with-an-operator' mechanism of TRAP transporters.

## eLife assessment

This article presents a cryo-EM structure of a tripartite ATP-independent periplasmic (TRAP) transporter that contributes to *Haemophilus influenzae* virulence. **Convincing** biophysical and cryo-EM experiments yield a **valuable** molecular model, but the functional importance of some of the molecular features identified remains to be demonstrated.

## Introduction

Secondary-active transporters are found in all domains of life (*Ren and Paulsen, 2005*; *Pao et al., 1998*; *Reizer et al., 1994*). They catalyse the movement of molecules across membranes, exploiting the free energy associated with electrochemical ion gradients to drive transport (see *Drew et al., 2021*; *Bosshart and Fotiadis, 2019* for recent reviews on the topic). In bacteria, secondary-active transporters are used for both the export of toxic compounds and for the import of nutrients needed for cell growth (*Drew et al., 2021*; *Henriquez et al., 2021*).

The import of carbohydrates across the plasma membrane is a key process for bacteria, particularly those pathogenic bacteria that adopt a scavenging lifestyle within their host (*Vimr, 2013*). For *Haemophilus influenzae*, a Gram-negative opportunistic pathogen, the ability to uptake host-derived sialic acids is important for pathogenesis (*Apicella, 2012*). Sialic acids are a diverse family of nine-carbon carbohydrates, and in humans, sialic acids are highly abundant in the respiratory and gastrointestinal tracts where they coat glycoconjugates as terminal sugars (*Vimr, 2013*; *Cohen and Varki, 2010*). The most common sialic acid found in humans is *N*-acetylneuraminate (Neu5Ac) (*Vimr, 2013*). To gain a growth advantage in these environments, bacteria such as *H. influenzae* evolved the ability to utilise host-derived sialic acid as a nutrient source (*Figure 1*). The sialic acid catabolic pathway provides an alternative source of carbon, nitrogen, and energy, and has been identified in 452 bacterial species, most of which are mammalian pathogens or commensals (*McDonald et al., 2016*). Alongside degradation, and perhaps more importantly, *H. influenzae* uses sialic acids to coat its lipopolysaccharide surface, and this sialylation in turn provides camouflage and protection from the human immune response (*Severi et al., 2005*; *Allen et al., 2005*). *H. influenzae* lacks the de novo sialic acid biosynthetic pathway, so sialic acids must be scavenged and transported into the cell (*Vimr et al., 2000*). Engineered *H. influenzae* strains that are unable to transport sialic acids into the cell have decreased virulence in animal models—evidence that the sialic acid transport pathway is a viable therapeutic target (*Jenkins et al., 2010*) and others have already developed inhibitors of sialic acid transporters (*Bozzola et al., 2022*). The sole sialic acid transporter in *H. influenzae* belongs to the tripartite ATP-independent periplasmic (TRAP) transporter family.

TRAP transporters are a major class of secondary-active transporters found only in bacteria and archaea (*Kelly and Thomas, 2001*; *Rosa et al., 2018*). They use energetically favourable cation gradients to drive the import of specific carboxylate-containing nutrients against their concentration-gradient, including C$_4$-dicarboxylates, α-keto acids, aromatic substrates, amino acids, and sialic acids (*Vetting et al., 2015*). A functional TRAP system is made up of a soluble substrate-binding 'P-subunit', and a membrane-bound complex comprising a small 'Q-subunit' and a large 'M-subunit'. For most TRAP transporters, the Q- and M-subunits are separate polypeptides (*Mulligan et al., 2011*; *Mulligan et al., 2012*), but in ~25% of sequences (InterPro analysis; *Blum et al., 2021*) the Q- and M-subunit polypeptides are fused into a single polypeptide (*Kelly and Thomas, 2001*). TRAP transporters are different from almost all other secondary-active transporters in that they can only accept substrates from the P-subunit (*Scheepers et al., 2016*). Analogous to ABC importers, the P-subunit is secreted

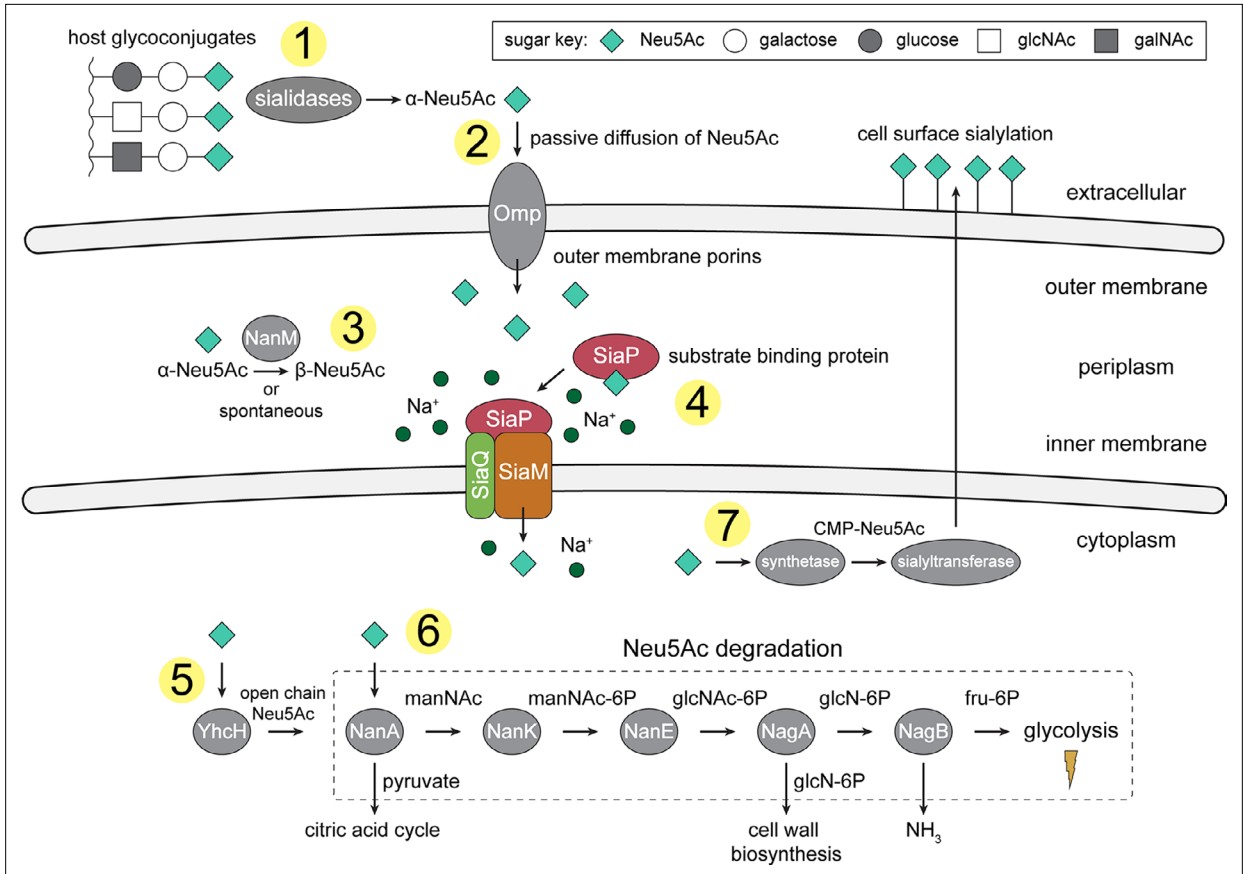

**Figure 1.** An overview of Neu5Ac metabolism in *H. influenzae*. (1) *H. influenzae* is sialidase negative and relies on environmental sialidases to hydrolyse and release terminal Neu5Ac from human glycoconjugates. (2) Outer membrane porins facilitate diffusion of Neu5Ac into the periplasm. (3) A mutarotase, NanM, catalyses the formation of β-Neu5Ac from α-Neu5Ac to prepare for active transport across the inner membrane. (4) Neu5Ac is captured by the high-affinity substrate-binding protein, SiaP. SiaP delivers Neu5Ac to the SiaQM TRAP transporter, which uses a Na$^+$ electrochemical gradient to drive transport. *H. influenzae* cannot synthesise Neu5Ac and relies solely on SiaPQM for obtaining environmental Neu5Ac. (5) Cytoplasmic processing of Neu5Ac by an anomerase, YhcH, generates the unfavourable open chain form in preparation for use by the first enzyme of the Neu5Ac degradation pathway, NanA. (6) Neu5Ac is sequentially degraded into cell wall constituents or fructose-6-phosphate, which can enter glycolysis. Five conserved enzymes (an aldolase, NanA; kinase, NanK; epimerase, NanE; deacetylase, NagA; and deaminase, NagB) are involved in this pathway which provides *H. influenzae* with carbon, nitrogen, and energy. (7) Alternatively, Neu5Ac can be activated by cytidine monophosphate and a sialic acid synthetase and added to lipooligosaccharides by a sialyltransferase. Neu5Ac, *N*-acetylneuraminate; manNAc, *N*-acetylmannosamine; manNAc-6P, *N*-acetylmannosamine-6-phosphate; glcNAc, *N*-acetylglucosamine; glcNAc-6P, *N*-acetylglucosamine-6-phosphate; glcN-6P, glucosamine-6-phosphate; fru-6P, fructose-6-phosphate; galNAc, *N*-acetylgalactosamine; CMP, cytidine monophosphate; Omp, outer membrane porin; NanM, Neu5Ac mutarotase; YhcH, Neu5Ac anomerase; NanA, Neu5Ac lyase; NanK, manNAc kinase; NanE, manNAc-6P epimerase; NagA, glcNAc-6P deacetylase; NagB, glcN-6P deaminase.

into the periplasm to capture host-derived substrates with high affinity and specificity. The substrate-loaded P-subunit subsequently delivers the substrate to the membrane transporter (Q- and M-subunits) (*Vetting et al., 2015*; *Müller et al., 2006*; *Johnston et al., 2008*; *Gangi Setty et al., 2014*).

The SiaPQM system from *H. influenzae* has fused Q- and M-subunits and is one of the best characterised TRAP systems to date (*Severi et al., 2005*; *Müller et al., 2006*; *Johnston et al., 2008*; *Fischer et al., 2015*; *Peter et al., 2022*), with key functional and biophysical studies informing a putative transport mechanism driven by a Na$^+$ gradient (*Mulligan et al., 2009*). Before transport into the cell by SiaPQM, sialidases release sialic acids from the terminal position of host glycoconjugates (*Figure 1*). *H. influenzae* is, however, sialidase negative and relies on the sialidases of other bacteria to generate free sialic acids (*Lichtensteiger and Vimr, 1997*). In Gram-negative bacteria, sialic acids diffuse into the periplasm through an outer membrane porin, such as the general porins OmpF or OmpC in many bacteria, Omp P2 in *H. influenzae* (*Vachon et al., 1985*; *Munson et al., 1992*), or by using a Neu5Ac specific porin such as NanC (*Wirth et al., 2009*). The substrate-binding protein SiaP

(P-subunit) binds Neu5Ac in the periplasm with high-affinity, undergoing a ligand-induced conformational change to a closed state (*Müller et al., 2006*; *Johnston et al., 2008*; *Gangi Setty et al., 2014*; *Fischer et al., 2015*). It is this closed form that then delivers Neu5Ac to the membrane bound Q- and M-subunits. This protein–protein interaction and the subsequent transport of Neu5Ac through the transmembrane subunits are not yet characterised. In particular, the conformational changes that take place for both SiaP and SiaQM to allow alternating access of the transporter have not been fully elucidated, though we have suggested previously that the binding of the closed P-subunit must induce either a global elevator-type motion or a local gating rearrangement as a part of an elevator mechanism (*Davies et al., 2023*). These changes allow the transporter to bind Neu5Ac and Na⁺ at the periplasmic side and transport them into the cytoplasm.

Here, we report the high-resolution cryo-electron microscopy (cryo-EM) structure of *Hi*SiaQM in amphipol and demonstrate that in detergent, amphipol, and a nanodisc environment, the fused *H. influenzae* SiaQM membrane subunits can stably exist as both monomeric (QM) and homodimeric (2QM) states. We determined the homodimeric (2QM) structure of *Hi*SiaQM using single-particle cryo-electron microscopy, the first structure of a fused TRAP transporter at near atomic resolution (2.99 Å, with local resolution extending to ~2.2 Å), which reveals new details of the transport mechanism. The structure, combined with functional and biophysical data, supports the hypothesis that *Hi*SiaQM operates using the 'elevator-with-an-operator' mechanism that we proposed for TRAP transporters (*Davies et al., 2023*).

## Results

### The cryo-EM structure of *Hi*SiaQM reveals a dimeric configuration

We assessed the stability of *Hi*SiaQM in several detergents that solubilise *Hi*SiaQM from *Escherichia coli* membranes. Lauryl maltose neopentyl glycol (L-MNG) led to more stable protein preparations compared to dodecyl-β-D-maltoside (DDM) (*Figure 2—figure supplement 1a*), which has been used previously (*Mulligan et al., 2012*; *Peter et al., 2022*). Curiously, *Hi*SiaQM purified with a high L-MNG concentration elutes as a major peak at 65 mL with a significant shoulder at 58 mL, and this elution profile is reversed at a low L-MNG concentration (*Figure 2—figure supplement 1a*)—a signature of a self-association, suggesting *Hi*SiaQM forms higher order oligomers. By comparison, the non-fused monomeric *Photobacterium profundum* SiaQM (*Pp*SiaQM) does not show this behaviour when expressed with the same purification tag and purified in the same low concentration of L-MNG, eluting at 67 mL, which we have previously shown to correspond to a monomeric species (*Davies et al., 2023*; *Figure 2—figure supplement 1a*).

We decided to purify *Hi*SiaQM using L-MNG under conditions that favour the larger species for the following reasons: (1) we were previously unsuccessful at determining the cryo-EM structure of the monomer because of the small size of the protein (72 kDa) and lack of features outside of the micelle or nanodisc that makes particle alignment difficult; (2) even with a megabody bound to increase the size and provide asymmetric features, other structural investigation on monomeric *Hi*SiaQM led to only moderate resolution data, which may reflect some instability or heterogeneity in the sample; and (3) we reasoned that avoiding fiducial markers such as nanobodies and megabodies would also allow exploration of the conformational dynamics of the transporter without any binder-associated bias. Thus, the larger *Hi*SiaQM species was isolated by exchanging purified protein out of L-MNG detergent and into amphipol A8-35, an amphipathic polymer used to solubilise and stabilise membrane proteins.

*Hi*SiaQM comprises two distinct dimeric configurations in approximately equal quantities (36%) in the 14,281-image dataset (*Figure 2a*, *Figure 2—figure supplement 2*, *Supplementary file 1*). Dimeric *Hi*SiaQM exists either as a side-by-side parallel dimer where both monomers face the same way, or an antiparallel dimer where one monomer is rotated around the dimer interface 180° relative to the other (*Figure 2a*, *Figure 2—figure supplement 2*). Dimerisation is mediated by the same interface in both of these structures but given that *Hi*SiaP is found in the periplasm and SiaQM requires SiaP to function, the parallel dimer is almost certainly the physiologically relevant conformation. The maps are of high resolution (overall 2.99 Å for the antiparallel dimer and 3.36 Å for the parallel dimer, FSC = 0.143 criterion) with both extending to 2.2 Å at the core, allowing for accurate fitting of helices and (almost) all side chains (*Figure 2—figure supplement 3*). The dimeric particles allowed for

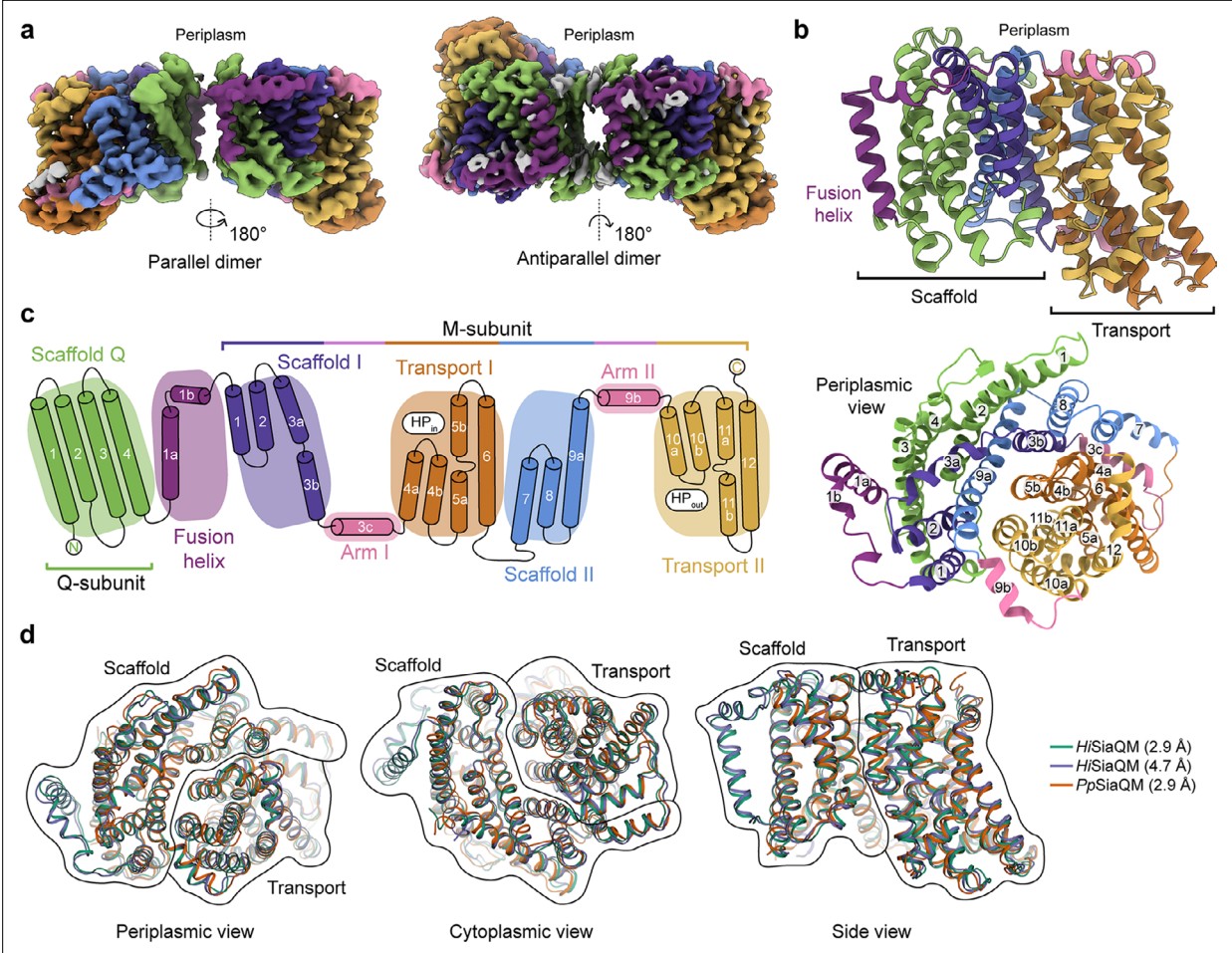

**Figure 2.** The structure of *Hi*SiaQM. (**a**) Coulomb maps for the parallel (3.36 Å) and antiparallel (2.99 Å) *Hi*SiaQM homodimers. The periplasmic surfaces of the monomers are facing the same direction for the parallel dimer (PDB: 8THI), whereas the periplasmic surface of one monomer is rotated 180° for the antiparallel dimer (PDB: 8THJ). The transport domain (orange and gold) is in the 'elevator down' conformation in all four monomers. The dimeric interface in both maps is distanced and neither has significant protein–protein interactions. The maps are coloured according to the topology in (**c**). Density consistent with phospholipids is coloured grey and is particularly present in the dimer interface of the higher resolution antiparallel dimer map. (**b**) Structural model of the *Hi*SiaQM monomer. The transport domain is in the 'elevator down' conformation with the substrate-binding site facing the cytoplasm. (**c**) The topology of *Hi*SiaQM is the same as the non-fused *Pp*SiaQM with the addition of the fusion helix. The M-subunit forms the transport domain (orange and gold) and bracing arm helices (pink) as well as a large portion of the scaffold (purple and blue). The Q-subunit is entirely used as a scaffold for the elevator transport mechanism. The fusion helix (purple) connects the scaffold and adds to its size. It also forms a short horizontal helix, similar to the arm helices of the M-subunit. (**d**) A structural overlay of *Hi*SiaQM (2.9 Å structure, green; 4.7 Å structure, purple) and *Pp*SiaQM (2.9 Å structure, orange) shows that the helices of the structures are well aligned, and all three structures are in the same conformation.

The online version of this article includes the following source data and figure supplement(s) for figure 2:

**Figure supplement 1.** Size-exclusion chromatography traces of SiaQM transporters suggests that *Hi*SiaQM exists as multiple species in detergent.

**Figure supplement 1—source data 1.** Original files for the gel and western blot analysis in *Figure 2—figure supplement 1b and c*.

**Figure supplement 1—source data 2.** Image containing *Figure 2—figure supplement 1b and c* and original files for the gel and western blot analysis with highlighted bands and sample labels.

**Figure supplement 2.** Cryo-electron microscopy (cryo-EM) workflow for structure determination.

**Figure supplement 3.** Representative density of the helices of *Hi*SiaQM.

structure determination without a protein fiducial marker bound (e.g., nanobody or antigen-binding fragment), which was required to solve the structure of the monomeric *Pp*SiaQM (**Davies et al., 2023**) and the lower resolution monomeric *Hi*SiaQM (**Peter et al., 2022**).

In all four *Hi*SiaQM protomers, the transport domain is in the inward-open conformation ('elevator down'), which is the substrate release state (r.m.s.d. for alignment = ~0.2 Å, **Figure 2a and b**). This is the

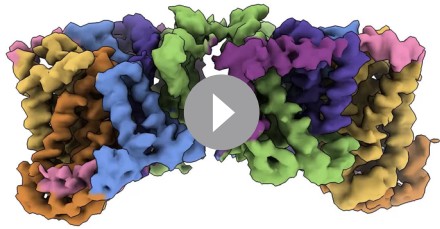

**Video 1.** Three-dimensional variability analysis of our *Hi*SiaQM reconstructions shows only very subtle motion at the dimer interface and does not show any global elevator motions in the final reconstruction. https://elifesciences.org/articles/92307/figures#video1

same conformation as the previous megabody-bound *Pp*SiaQM structures in amphipol and a nanodisc (*Davies et al., 2023*) and also the same as the recently reported lower resolution (4.7 Å) megabody-bound *Hi*SiaQM structure in a nanodisc (*Peter et al., 2022*). Three-dimensional variability analysis (3DVA) (*Punjani and Fleet, 2021*) of our *Hi*SiaQM reconstructions shows only very subtle motion at the dimer interface and does not show any global elevator motions in the final reconstruction (*Video 1*). That we observe the inward-open conformation without either a bound P-subunit or fiducial marker suggests that this is the resting state of the transporter under experimental conditions (in the absence of a membrane bilayer, membrane potential, and chemical gradients).

Both dimers form through an interface between the Q-subunits of the monomers. The relatively flat sides of the Q-subunits are positioned adjacent to each other in an interface that is not intimately connected by side-chain interactions (*Figure 2a*). When refined using dimeric masks, the parallel dimer has slightly lower resolution at the dimer interface (scaffold region) compared to the antiparallel dimer. Local refinement of each monomer within the parallel dimer improved the resolution at the scaffold region, suggesting this is perhaps a relatively dynamic interface. The buried interface area is small (~200 Å²) for the antiparallel configuration, and even smaller (~100 Å²) for the parallel configuration (*Krissinel and Henrick, 2007*). Further still, the parallel interface area is formed by a small number of residues at the N-termini, which have modest resolution. The majority of the interface for these dimers must therefore be mediated by lipid molecules that bridge between the monomers, some of which we observe directly in the cryo-EM maps (discussed later). Small interfaces have been observed in the elevator-type $Na^+/H^+$ antiporter NhaA from *E. coli* (*Gupta et al., 2017*) and throughout the SLC26 family (*Chang et al., 2019*) (also elevator-type), but the *Hi*SiaQM interfaces are much smaller than other dimeric ion transporter superfamily members, such as the TRAP-related symporter VcINDY, a divalent anion/$Na^+$ symporter (DASS) from *Vibrio cholerae* (*Mulligan et al., 2016*; *Mancusso et al., 2012*; *Nie et al., 2017*), and the antimicrobial efflux pumps MtrF (*Su et al., 2015*) and YdaH (*Bolla et al., 2015*), which are all elevator-type homodimers with buried interface areas of ~1800–2100 Å².

The topology of *Hi*SiaQM is similar to the non-fused *Pp*SiaQM, aside from the additional fusion helix that links the Q- and M-subunits across the membrane (*Figure 2c*). The helix starts on the cytoplasmic side, packing against the Q-subunit and crossing the membrane vertically, then turns 90° and runs laterally (perpendicular to the membrane) before forming a loop and connecting to the M-subunit on the periplasmic side (*Figure 2a–c*). An overlay with the recently determined SiaQM structures shows that the main chains are similar (*Pp*SiaQM to high-resolution *Hi*SiaQM r.m.s.d. = 1.2 Å across 520 Cα-atoms and lower resolution *Hi*SiaQM to high-resolution *Hi*SiaQM r.m.s.d. = 1.0 Å across 571 Cα-atoms, *Figure 2d*). All three structures are in the 'elevator down' conformation with close alignment between the 16–17 helices (*Figure 2d*). Differences in the loops of the proteins exist, including the positioning of the 'clamshell' loops of the $Na^+$-binding sites (discussed later).

## *Hi*SiaQM exists in a stable monomer–dimer self-association

The oligomeric assembly of TRAP transporters within the membrane has implications for the transport cycle, as well as the interaction between SiaP and SiaQM. Secondary-active transporters can exist as monomers (*Kaback, 2005*; *Faham et al., 2008*), or as larger multimeric complexes of oligomers (*Perez et al., 2011*; *Yernool et al., 2004*). For example, VcINDY (*Mulligan et al., 2016*; *Mancusso*

*et al., 2012*; *Nie et al., 2017*), MtrF (*Su et al., 2015*), and YdaH (*Bolla et al., 2015*) all share the same fold and have a dimeric structure (*Vergara-Jaque et al., 2015*). These elevator-type transporters form oligomers at the scaffold domains, which is thought to provide the stability needed for elevator movements of the transport domain. Recently, a TRAP-related fused transporter belonging to the TAXI (TRAP-Associated eXtracytoplasmic Immunogenic proteins) TRAP subfamily from *Proteus mirabilis* also eluted as two peaks during size-exclusion chromatography, signalling a likely self-association (*Roden et al., 2023*). In contrast, the non-fused TRAP transporter *Pp*SiaQM uniquely functions as a monomer (1QM) (*Davies et al., 2023*), where instead of homodimerisation, the scaffold is formed by heterodimerisation of the Q- and M-subunits. Recent work on *Hi*SiaQM also proposes a monomeric (1QM) functional transporter (*Mulligan et al., 2012*; *Peter et al., 2022*). However, our high-resolution dimeric cryo-electron microscopy structures and the self-association behaviour observed from size-exclusion chromatography (*Figure 2—figure supplement 1a*) infer a stable dimeric architecture for *Hi*SiaQM is possible.

To further support this result, *Hi*SiaQM purified in L-MNG and DDM was characterised by sedimentation velocity analytical ultracentrifugation (SV-AUC), resulting in two distinct species in each of the samples (*Figure 3a*, *Figure 3—figure supplements 1–3*). As there were no contaminants following purification (*Figure 2—figure supplement 1b*), the two species represent a protein self-association, consistent with that observed during purification. The oligomeric state of the two species was verified using SV-AUC analysis that compares the measured masses with theoretical masses. This suggests that *Hi*SiaQM is present as a monomer and a dimer in both L-MNG (*Figure 3a*) and DDM (*Figure 3—figure supplement 1*). The two species have sedimentation coefficients (all are corrected for water and 20°C unless otherwise specified) of 7.3S and 9.9S in L-MNG, and 7.6S and 10.3S in DDM. The calculated frictional ratios for the monomeric species are 1.2 in L-MNG and 1.1 in DDM. These are consistent with a globular particle, as we would expect for a protein in a detergent micelle. The calculated frictional ratios for the dimeric species are 1.2 in L-MNG and 1.0 in DDM, which are also consistent with a globular particle.

We then solubilised *Hi*SiaQM in amphipol (A8-35) and nanodiscs (cNW11, ~11 nm in diameter *Nasr et al., 2017*) to ascertain whether these systems might afford better stability and again found two species consistent with a monomer and a dimer. SV-AUC analysis of amphipol-solubilised protein (*Figure 3b*) is consistent with a monomer at 5.9S and a dimer at 8.3S, when considering the oligomeric information obtained from SV-AUC in L-MNG (*Figure 3a*). In nanodiscs, a reconstitution ratio of 1:4:80 of *Hi*SiaQM:MSP:lipid (MSP = membrane scaffold protein) resulted in multiple peaks during size-exclusion chromatography (*Figure 3—figure supplement 4a*). Three fractions across the elution profile were analysed with SV-AUC and the main species in each had sedimentation coefficients of 4.0S, 6.8S, and 9.0S (*Figure 3—figure supplement 4b*). The species at 4.0S is most consistent with empty nanodiscs, and the species at 6.8S and 9.0S are most consistent with nanodiscs containing monomeric and dimeric *Hi*SiaQM, respectively (*Figure 3—figure supplement 4b*).

To summarise, we have demonstrated that *Hi*SiaQM can exist in a monomer–dimer state in two detergents, amphipol and nanodiscs, which mimic the native environment. The proposal that *Hi*SiaQM forms dimers within the plasma membrane is supported by the observation that a stable dimer with low aggregation propensity forms when solubilised in L-MNG (*Figure 2—figure supplement 1a*, blue trace). Stable species also exist when solubilised in amphipol (*Figure 3b*), particularly evidenced by the collection of cryo-EM data to <3 Å resolution. Under the same L-MNG detergent conditions, similar concentrations, and with an identical $Ni^{2+}$ affinity tag, non-fused *Pp*SiaQM does not show this behaviour (*Figure 2—figure supplement 1a*)—evidence that dimerisation is a result of a self-association and not the $Ni^{2+}$ affinity tag or cohabitation, where the number of available micelles is low enough that two proteins cohabitate the same micelle for stable solubilisation. The dimeric assembly of *Hi*SiaQM is consistent with other examples of transporters that use the elevator-type mechanism (e.g., VcINDY; *Mulligan et al., 2016*; *Mancusso et al., 2012*; *Nie et al., 2017*), but clearly different from that reported recently for *Pp*SiaQM and *Hi*SiaQM TRAP transporters (*Peter et al., 2022*; *Davies et al., 2023*).

## Lipid-binding sites

As a transporter, *Hi*SiaQM has evolved alongside lipids, which are increasingly shown to be important for general transporter function and oligomerisation (*Nji et al., 2018*; *Landreh et al., 2017*; *Matsuoka*

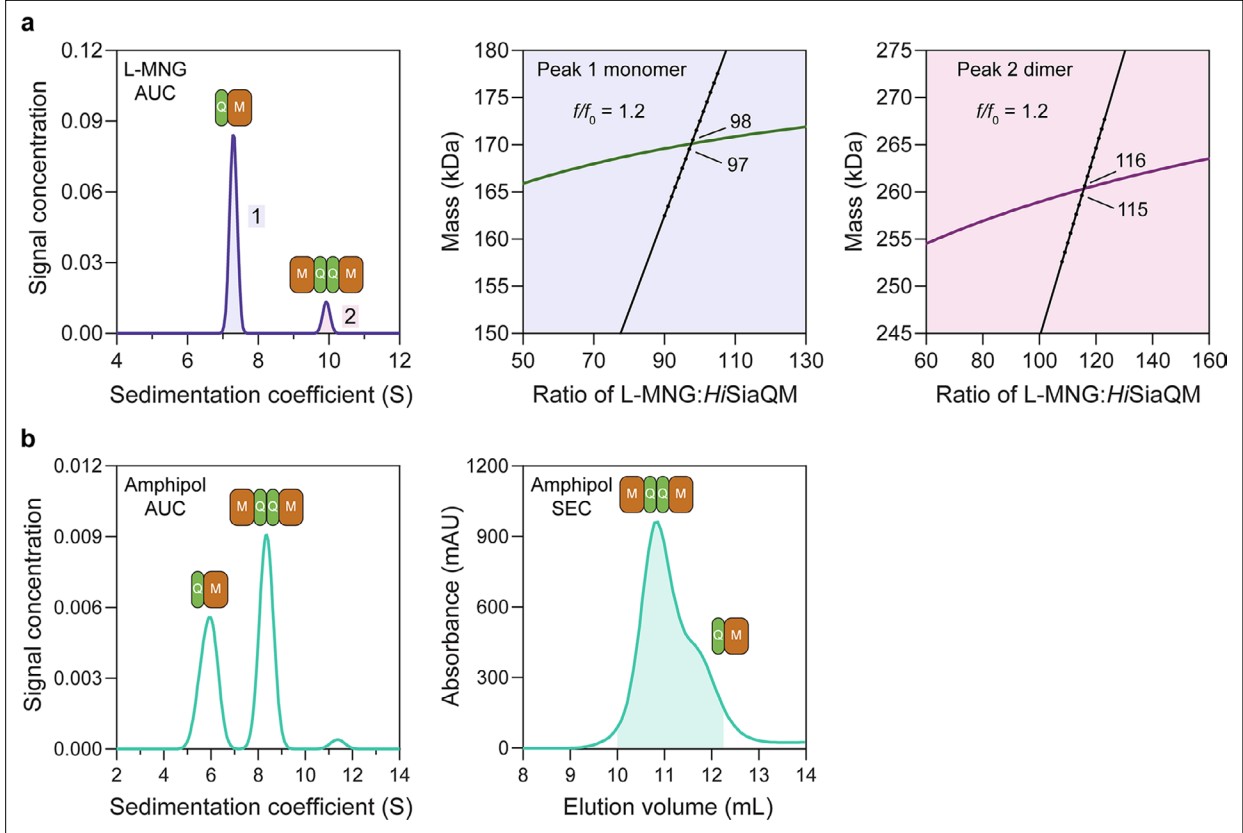

**Figure 3.** *Hi*SiaQM self-association in lauryl maltose neopentyl glycol (L-MNG) and amphipol. (**a**) Sedimentation velocity analytical ultracentrifugation (SV-AUC) analysis of *Hi*SiaQM in L-MNG (left panel). Two well-resolved species exist at 7.3S (diffusion coefficient, D = $4.8 \times 10^{-7}$ cm$^2$/s) and 9.9S (D = $4.2 \times 10^{-7}$ cm$^2$/s), with the larger peak constituting 85% of the signal. The species at 7.3S (peak 1, blue shading) is most consistent with *Hi*SiaQM as a monomer with ~98 molecules of L-MNG bound (middle panel; green = measured mass, black = theoretical mass), calculated from the experimental sedimentation and diffusion coefficients. These calculations suggest that peak 1 existing as a dimer is unlikely, as the dimeric protein would only have ~14 molecules of L-MNG bound. Additionally, the calculated $f/f_0$ of a monomer for peak 1 is 1.2, consistent with a protein in a detergent micelle. The species at 9.9S (peak 2, pink shading) is most consistent with *Hi*SiaQM as a dimer with ~116 molecules of L-MNG bound (calculated) (right panel; purple = measured mass, black = theoretical mass); peak 2 existing as a monomer is not possible, as the protein clearly has a smaller species in peak 1 and cannot be divided further than a monomer, and a trimer is also unlikely as the trimeric protein would only have ~32 molecules of L-MNG bound (calculated). Additionally, the calculated $f/f_0$ of a dimer for peak 2 is also 1.2, again consistent with a protein in a detergent micelle. These calculations do not account for bound lipid molecules. (**b**) Left panel: SV-AUC analysis of amphipol-solubilised *Hi*SiaQM (initially purified in L-MNG) shows two distinct species present at 5.9S and 8.3S. These are monomeric and dimeric species as L-MNG-solubilised protein exists as these oligomeric states at 7.3S and 9.9S as in (**a**). Right panel: representative size-exclusion chromatogram of amphipol-solubilised *Hi*SiaQM favouring the dimeric state. The main peak at ~10.8 mL contains dimeric *Hi*SiaQM and the shoulder at ~11.8 mL contains monomeric *Hi*SiaQM. The sample used for structure determination is shaded turquoise.

The online version of this article includes the following figure supplement(s) for figure 3:

**Figure supplement 1.** *Hi*SiaQM self-association in dodecyl-β-d-maltoside (DDM).

**Figure supplement 2.** *Hi*SiaQM self-association in lauryl maltose neopentyl glycol (L-MNG) (interference analysis).

**Figure supplement 3.** *Hi*SiaQM self-association in dodecyl-β-D-maltoside (DDM) (interference analysis).

**Figure supplement 4.** Solubilisation of *Hi*SiaQM in nanodiscs.

*et al., 2022*). Evident in our cryo-EM maps are well-defined phospholipid densities associated with areas of *Hi*SiaQM that may be important for the function of an elevator-type mechanism (***Figure 4***), but require further testing. The two most compelling areas are the dimer interface and a lipid-binding pocket formed by the fusion helix.

There is a void region at the dimer interface in both structures that likely contains amphipol polymers and lipids. Indeed, we observe density for lipids in this region (best seen in the slightly higher resolution antiparallel dimer) (***Figures 2a*** and ***Figure 4b***). Notably, these densities are adjacent to tryptophan residues (***Figure 4—figure supplements 1 and 2***), known for their role in anchoring

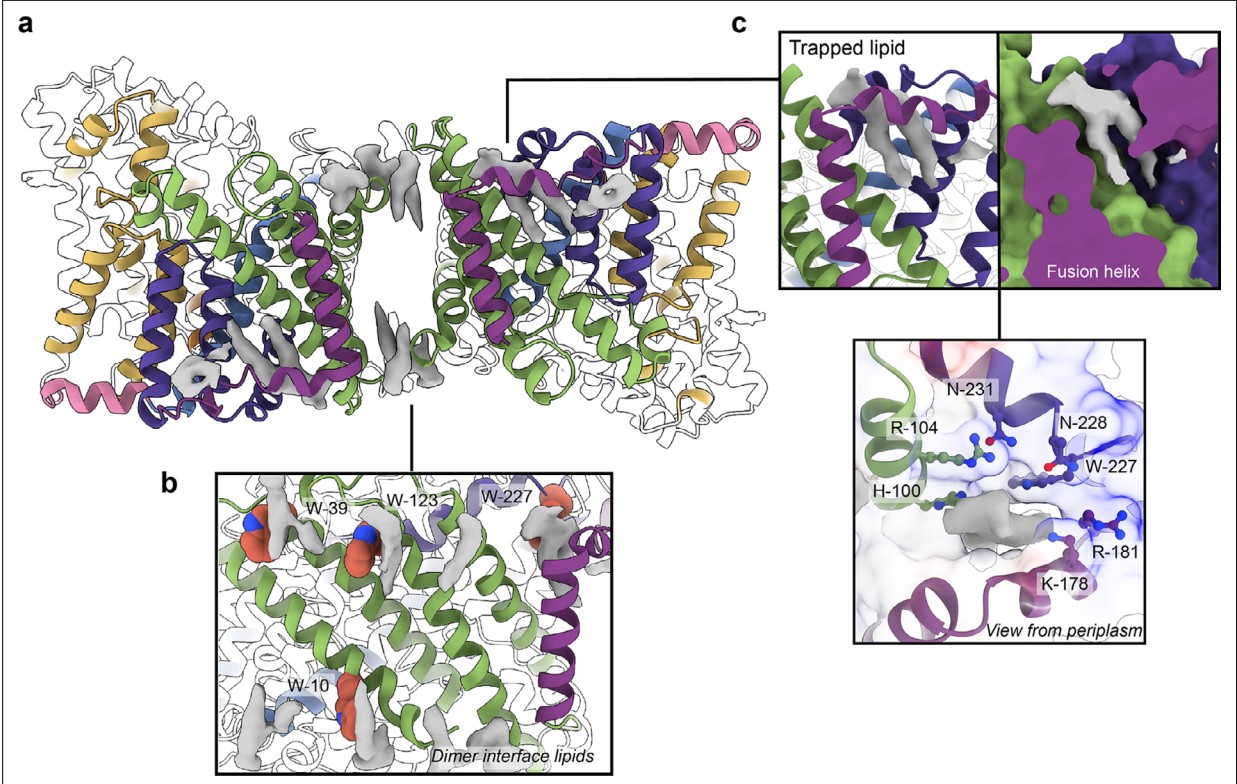

**Figure 4.** Phospholipids bound to *Hi*SiaQM. (**a**) The dimer has well-defined areas of density (grey) that correspond to bound phospholipids. Two mechanistically important areas are the dimer interface and fusion helix pocket. (**b**) Phospholipids are present at the dimer interface, including in close contact with the anchoring tryptophan residues (shown as spheres) of the Q-subunits that provide stability to the scaffold domain. (**c**) A single phospholipid is trapped in a pocket formed by the fusion helix (protein model surface shown in colour, EM density in grey). The lipid is on the periplasmic side of the transporter and the headgroup appears to be coordinated by residues surrounding the top of the pocket, which have a generally positive character.

The online version of this article includes the following figure supplement(s) for figure 4:

**Figure supplement 1.** Protein sequence alignment of tripartite ATP-independent periplasmic (TRAP) transporter QM-subunits.

**Figure supplement 2.** Protein sequence alignment of tripartite ATP-independent periplasmic (TRAP) transporter QM-subunits.

membrane proteins (*de Jesus and Allen, 2013*), and one of the positions is where we also observe clear lipid density in the *Pp*SiaQM structure in a nanodisc (*Davies et al., 2023*). The discovery of these annular lipids is surprising given that the protein was first extracted and extensively purified in L-MNG detergent, followed by exchange into A8-35 amphipol for cryo-EM studies. There are some weaker densities around the periphery of the protein that likely correspond to amphipol, but we can confidently identify densities for phospholipids given their distinctive hairpin shape (*Figure 4*). That lipids have remained bound in this area through the isolation and purification supports that these are strong interactions, which thereby anchor the scaffold domain in the membrane. The lipids at this interface may also have a role in strengthening dimerisation, as seen in other secondary-active transporters such as NhaA (*Gupta et al., 2017*; *Landreh et al., 2017*).

The architecture of the fusion helix leaves a pocket between the lateral part of the helix and the main bundle of the transporter. A closer inspection of the map density shows that a distinct hairpin-shaped phospholipid density is positioned in this pocket (*Figure 4c*). This enclosed pocket appears to tightly fit only one phospholipid, which is presumably trapped here during protein folding. Several residues from the scaffold portion are well-poised to interact with a lipid headgroup, including H100, R104, K178, and W227 (*Figure 4c*). Analysis of the electrostatic surface of the pocket shows a significant amount of positive charge. We suggest that this site likely binds a negatively charged phospholipid and have modelled phosphatidylglycerol into the density. The predicted interacting residues are

not well conserved amongst fused TRAP transporters (*Figure 4—figure supplements 1 and 2*), which is possibly a reflection of varying bacterial membrane compositions.

This site and the presence of a lipid were not identified in the recent structure of *Hi*SiaQM in a nanodisc (containing only the zwitterionic lipid DMPC)—instead, amino acids are modelled to fill the hydrophobic pocket. This is notable, as *Peter et al., 2022* hypothesised that W227 is important for the SiaP:SiaQM interaction, and show that mutation of this residue to an arginine significantly affects bacterial growth in an in vivo growth assay. The resolution of our structures allows us to unambiguously model this residue facing away from the surface and engaged in an interaction with the trapped phospholipid, further supporting our identification of tryptophan residues in anchoring roles in the scaffold of TRAP transporters (*Davies et al., 2023*). This finding is important in the context of the transport mechanism used by *Hi*SiaPQM. As the identified phospholipid is intercalated between scaffolding helices and is very close to the putative SiaP:SiaQM interface, it is feasible that this site is physiologically significant and may influence the stability of the transporter or even the conformational dynamics of the entire system. The fusion helix and concomitant lipid molecule may provide a more structurally rigid scaffold than a Q-M heterodimer, such as *Pp*SiaQM, although how this impacts the elevator transition requires further testing. While this binding pocket is likely found in a number of fused TRAPs (based on sequence predictions, e.g., *Fn*SiaQM and *Aa*SiaQM in *Figure 4—figure supplements 1 and 2*), it is not clear whether they also bind lipids here without experimental data. We emphasise that this phospholipid binding pocket is not present in *Pp*SiaQM, which may point to differences in the dynamics of the scaffolds and how the scaffold interacts with SiaP in non-fused systems.

## Neu5Ac and Na$^+$-binding sites

Although others have already demonstrated Neu5Ac transport for the *Hi*SiaPQM system (*Mulligan et al., 2009*), we verified that our recombinant *Hi*SiaPQM is active after solubilisation in the previously untested detergent L-MNG. Time-dependent uptake of [$^3$H]-Neu5Ac was measured after reconstituting *Hi*SiaQM into potassium containing proteoliposomes (*Figure 5a*). This method allows for the transport dependence on a Na$^+$ gradient ($\Delta\mu$Na$^+$, generated by adding external Na$^+$) or a membrane potential ($\Psi$, generated by adding valinomycin) to be measured (*Wahlgren et al., 2018*).

Transport by *Hi*SiaQM was dependent on a Na$^+$ gradient ($\Delta\mu$Na$^+$) (*Figure 5a*, green and orange circles) and was stimulated threefold in the presence of a membrane potential ($\Psi$) (*Figure 5a*, green circles). *Hi*SiaP is required for transport since without external *Hi*SiaP transport was negligible. These results agree with previous work on *Hi*SiaQM purified in decyl-β-D-maltoside (DM) (*Mulligan et al., 2009*) and show that our tagged construct is active and not inactivated by L-MNG. Net transport by *Hi*SiaQM is electrogenic, as activity is enhanced when an inside negative membrane potential ($\Psi$) is imposed (*Figure 5a*). Since Neu5Ac has a single negative charge at neutral pH, electrogenic transport means two or more Na$^+$ ions are transported for every Neu5Ac imported. Varying external [Na$^+$] at a fixed substrate concentration resulted in a Hill coefficient of 2.9 (95% C.I. = 2.2–3.9), implying that at least two Na$^+$ ions are co-transported during each transport cycle (*Figure 5b*). These data are consistent with the previous work (*Mulligan et al., 2009*), as well as the stoichiometry we recently determined for *Pp*SiaQM (*Davies et al., 2023*).

The fused *Hi*SiaPQM system appears to have a higher transport activity than the non-fused *Pp*SiaPQM system. With the same experimental setup used for *Pp*SiaPQM (5 µM Neu5Ac, 50 µM SiaP) (*Davies et al., 2023*), the accumulation of [$^3$H]-Neu5Ac by the fused *Hi*SiaPQM is approximately tenfold greater. Although this difference may reflect the reconstitution efficiency of each proteoliposome preparation, it is possible that it has evolved as a result of the origins of each transporter system—*P. profundum* is a deep-sea bacterium and as such the transporter is required to be functional at low temperatures and high pressures. In contrast, *H. influenzae* is mesophilic, found in the human respiratory tract and therefore more likely to show native-like rates in ambient experimental conditions. A similar pattern has been reported for the elevator transporter Glt$_{Ph}$, where transporters from hyperthermophilic species are slow, and mesophilic versions are faster (*Huysmans et al., 2021*). These organisms also vary substantially in membrane lipid composition, which is currently too difficult to test in this experimental setup.

Next, we examined the *Hi*SiaQM Na$^+$-binding sites in our cryo-EM structure (*Figure 5c*). The higher resolution antiparallel dimer structure shows density for Na$^+$ ions at both Na1 and Na2 sites (*Figure 5—figure supplement 1a*), confirming the sites we identified and tested in *Pp*SiaQM (*Davies*

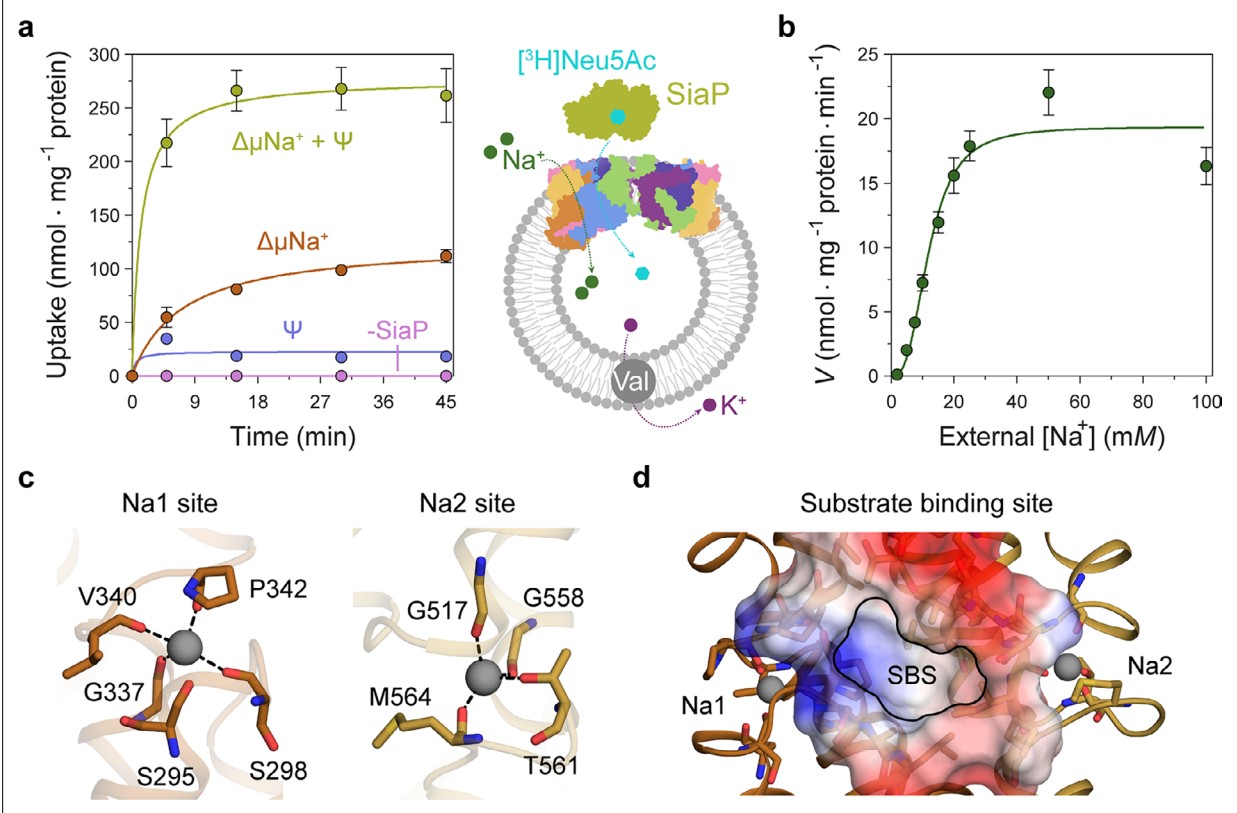

**Figure 5.** Transport assays demonstrate that lauryl maltose neopentyl glycol (L-MNG)-solubilised *Hi*SiaQM is functional. (**a**) [³H]-Neu5Ac uptake was measured at multiple time intervals under each condition and used to calculate transport rates. *Hi*SiaQM had the highest activity in the presence of *Hi*SiaP, a membrane potential and a Na⁺ gradient (green circles). Approximately one-third of this rate was present without a membrane potential (orange circles). Transport was low in the absence of Na⁺ (blue circles) and negligible without *Hi*SiaP (pink circles). Error bars represent the standard error of the mean (SEM) for three technical replicates, except without *Hi*SiaP, which has two replicates. The assay diagram contains the parallel *Hi*SiaQM structure, coloured by topology as in *Figure 2a and c*. This is for visual presentation; it is not known if the transporter exists as a dimeric species in the assay. (**b**) The rate of transport is dependent on the concentration of Na⁺, showing a sigmoidal response (Hill coefficient = 2.9 [95% C.I. 2.2–3.9]). The $K_M$ for Na⁺ is 12 mM (95% C.I. 10–14 mM). The displayed response shows that *Hi*SiaPQM operates close to its maximum measured rate at a reasonably low external Na⁺ concentration (25 mM). Error bars represent the SEM of five technical replicates. (**c**) Two Na⁺-binding sites (Na1 and Na2) were identified in *Hi*SiaQM. These sites share highly similar coordinating residues with *Pp*SiaQM. At the Na1 site, a Na⁺ ion (grey) is coordinated by the carbonyl groups of S298, G337, V340 and P342 (orange sticks, coordination shown as black dashes). S295 is also shown but its carbonyl is positioned just outside the coordination distance in our structure. At the Na2 site, a Na⁺ ion (grey) is coordinated by the carbonyl groups of G517, G558, and M564, and the side-chain hydroxyl of T561 (gold sticks, coordination shown as black dashes). (**d**) A putative substrate-binding site (SBS, outlined) is located in the transport domain of *Hi*SiaQM (orange and gold). The mostly hydrophobic binding site (shown as sticks and electrostatic surface) exists between the two Na⁺-binding sites and is large enough to bind Neu5Ac.

The online version of this article includes the following figure supplement(s) for figure 5:

**Figure supplement 1.** The Na⁺ and substrate-binding sites of *Hi*SiaQM.

*et al., 2023*), and also the aforementioned stoichiometry of Na⁺ ions transported determined in the *Hi*SiaQM proteoliposome assay (*Mulligan et al., 2009*). The Na1 and Na2 sites across all four monomers are highly similar, although subtly different to those reported previously (*Figure 5—figure supplement 1b*). The resolution of the structures allows for the unambiguous assignment of side-chain positions at these binding sites and enables mechanistically important residues to be distinguished.

At the Na1 site, coordination is achieved by the backbone carbonyls of S298, G337, V340, and P342 in a tetrahedral pattern, with a valence = 1, as determined by CheckMyMetal (*Zheng et al., 2014*; *Brown, 2009*). The backbone carbonyl of S295 is also positioned for coordination but is slightly further away at 2.9/3.0 Å. In contrast, the equivalent carbonyl in *Pp*SiaQM (S103) is closely associated with the Na⁺ ion to form a trigonal bipyramidal coordination pattern. Na2 coordination is achieved by the backbone carbonyls of G517, G558 and M564, and the side-chain hydroxyl of T561, which together coordinate the Na⁺ ion in a tetrahedral pattern with a valence = 1.2/1.3. In contrast to the

Na2 site of *Pp*SiaQM, the backbone carbonyl of T561 is positioned further away at 4.2 Å distance and does not directly coordinate the Na⁺ ion in our structure. In the *Pp*SiaQM structure, the equivalent residue (T369) coordinates the Na⁺ ion with both the side chain and main chain of T561 to form a square planar coordination pattern. S298, P342, G517, and G558 are highly conserved in TRAP transporters; S295, G337, and T561 sometimes vary as other small residues and V340 and M564 often vary as other hydrophobic side chains (*Figure 4—figure supplements 1 and 2*). S298 did not show an effect on growth when mutated to alanine previously (*Peter et al., 2022*) and therefore may not be considered mechanistically important, but our structure shows that it coordinates Na1 through the backbone carbonyl, masking the effect of the side-chain mutation.

Our structures have a large cytoplasm-facing solvent-accessible cavity housing the putative binding site for Neu5Ac that is flanked by the Na1 and Na2 Na⁺ ion sites (*Figure 5d*), similar to *Pp*SiaQM. The dataset was collected without Neu5Ac and not surprisingly, there is no ligand density in the binding sites. The putative Neu5Ac binding site is lined with residues S300, A301, L302, P338, S343, I344, A345, I348, D521, A522, L523, Q526, T552, M555, M556, I559, and M566 of the transport domain which form a largely hydrophobic binding site (*Figure 5d*). This site is large enough to bind Neu5Ac and is more defined than previously reported (*Figure 5—figure supplement 1c*; *Peter et al., 2022*). Residues D304 and D521 each have a significant effect on bacterial growth when mutated to alanine and were thought to be involved in Na⁺ ion or substrate binding (*Peter et al., 2022*). Our structure shows that neither are involved in Na⁺ binding and although D521 does line the substrate-binding pocket and introduces some charge to the site, D304 faces away from the substrate-binding site and instead coordinates the residues that bind the Na⁺ ion at the Na1 site.

## On the formation of the tripartite complex

The observation that *Hi*SiaQM can form a dimer has implications for the mechanism of transport, and as yet the affinity for the SiaP:SiaQM complex has not been directly established. To address this, we used analytical ultracentrifugation with fluorescence detection to determine the affinity between *Hi*SiaP and *Hi*SiaQM in the presence of Neu5Ac. Fluorescein isothiocyanate (FITC) was used to fluorescently label *Hi*SiaP, with the labelling conditions adjusted to provide just one FITC label per *Hi*SiaP protein. This allowed for a usable signal to be obtained at 10 nM *Hi*SiaP and minimised any inhibitory effects

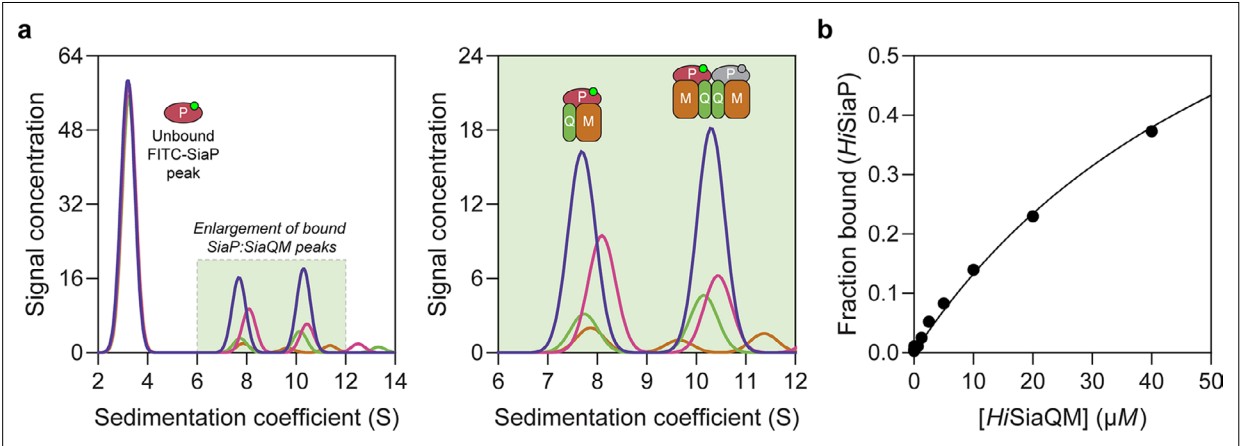

**Figure 6.** Sedimentation velocity analytical ultracentrifugation (AUC) analysis of the interaction between *Hi*SiaQM and *Hi*SiaP in lauryl maltose neopentyl glycol (L-MNG) detergent. (**a**) Titrating increasing concentrations of *Hi*SiaQM (blue, 40 µM [2.88 mg/mL]; pink, 20 µM; green, 10 µM; brown, 5 µM; other concentrations omitted for clarity) against fluorescently labelled FITC-*Hi*SiaP (10 nM) identifies a shift in the signal for *Hi*SiaP from 3S to ~7.5S and ~10.5S. This shift demonstrates binding to *Hi*SiaQM and identifies two bound species with different sedimentation coefficients. The two species are annotated with the most likely binding stoichiometries (one or two *Hi*SiaP monomers [red and grey] may be binding the dimeric species). (**b**) Fitting a binding model to the data (fraction bound = [P]$_{total}$/[P]$_{total}$ + K$_D$) estimates a K$_D$ of 65 µM (95% C.I. = 62–69 µM, R² = 0.99).

The online version of this article includes the following figure supplement(s) for figure 6:

**Figure supplement 1.** *Hi*SiaP is monomeric.

**Figure supplement 2.** Modelling the *Hi*SiaPQM complex.

**Figure supplement 3.** Residues involved in scaffold stabilisation and the interaction with SiaP.

on complex formation as 22 of the 31 lysine residues of *Hi*SiaP are spatially distant from the putative interacting surface. Titrating *Hi*SiaQM in L-MNG to 40 µM (a 4000-fold excess over *Hi*SiaP, 10 nM) in the presence of 5 mM Neu5Ac resulted in the development of two peaks at ~7.5S and ~10.5S (*Figure 6a*). Integration of the signal that shifted to a larger species resulted in ~40% of the *Hi*SiaP bound at the highest concentration of *Hi*SiaQM (40 µM). This experiment allowed us to estimate a $K_D$ for the complex of 65 µM (95% C.I. = 62–69 µM, *Figure 6b*). Avoiding non-ideality meant that we could only use concentrations of *Hi*SiaQM approaching the $K_D$, but the binding model fit was very good ($R^2 = 0.99$). This is in agreement with previous work that used surface plasmon resonance with biotin-immobilised *Hi*SiaQM in DDM, where a $K_D$ of ~1 µM was estimated, but it was stressed that the true $K_D$ is very likely weaker due to the inability to fit a 1:1 binding model (*Peter et al., 2022*), which may have been complicated by the presence of dimeric *Hi*SiaQM or aggregation.

Interestingly, the fluorescent signal shifted to two larger species. Based on the data for *Hi*SiaQM in L-MNG (*Figure 3a*), the two species at ~7.5S and ~10.5S likely correspond to monomeric *Hi*SiaQM with one *Hi*SiaP monomer bound and dimeric *Hi*SiaQM with either one or two *Hi*SiaP monomers bound (*Figure 6a*). To support this stoichiometry, we checked the in-solution oligomeric state of *Hi*SiaP, as there are examples of unusual TRAP substrate-binding proteins that form dimers (*Vetting et al., 2015*; *Gonin et al., 2007*). Both analytical ultracentrifugation (0.85 mg/mL, 25 µM) and small-angle X-ray scattering experiments (injected at 12.3 mg/mL, 360 µM) suggest that *Hi*SiaP is monomeric in solution (*Figure 6—figure supplement 1*).

To understand the observed complex formation, the interaction between *Hi*SiaP (PDB: 3B50) and *Hi*SiaQM (PDB: 8THI) was modelled using AlphaFold2 (*Jumper et al., 2021*). The predicted binding mode does allow two *Hi*SiaP proteins to bind the parallel *Hi*SiaQM dimer without significant steric clashes (*Figure 6—figure supplement 2*) and each monomer has the same binding mode as the *Pp*SiaPQM model (*Davies et al., 2023*).

The resolution of our structure enabled us to model the side chains and rationalise previous mutagenic experiments designed to define the SiaP:SiaQM interaction. We demonstrated that the periplasmic-facing residue R292 in *Pp*SiaM is important for TRAP transporter activity and hypothesised that this residue is important for both the interaction with SiaP and rigidifying the scaffold-lateral arm helix junction (*Davies et al., 2023*). This residue is strictly conserved in sialic acid TRAP transporters but is not conserved across the TRAP transporter family (*Figure 4—figure supplements 1 and 2*). Our structure shows that the equivalent residue in *Hi*SiaQM (R484) adopts a similar solvent-accessible conformation and participates in an ionic interaction with D480, which in turn is interacting with K236 (*Figure 6—figure supplement 3*). Together these residues form a salt bridge network between two helices of the scaffold of the transporter. Consistent with this, mutation to R484E resulted in decreased bacterial growth and reduced the binding of SiaP to SiaQM in vitro (*Peter et al., 2022*). Furthermore, mutations of residues on SiaP predicted to interact with this region reduce activity in

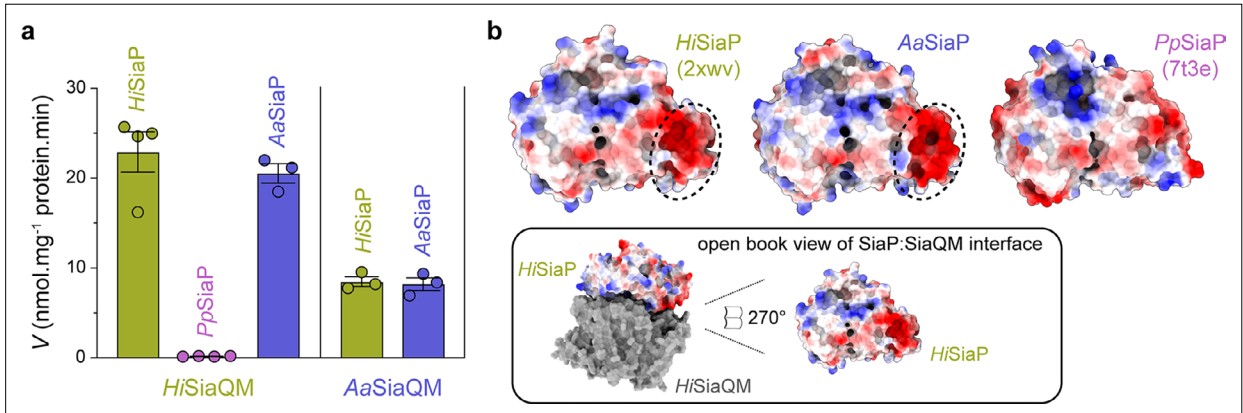

**Figure 7.** Subunit substitution transport assays. (**a**) Transport was measured with subunit substitution of *Hi*SiaPQM with the fused SiaPQM from *A. actinomycetemcomitans* (*Aa*) and the non-fused SiaPQM from *P. profundum* (*Pp*). Transport activity was measured in the presence of a membrane potential and a Na$^+$ gradient. The mean activity is shown as bars with SEM error from at least three technical replicates (n = 3 or 4). (**b**) Electrostatic surface comparison of the putative SiaQM interaction surfaces of *Hi*SiaP, *Aa*SiaP and *Pp*SiaP. The SiaP proteins of the two fused systems have a greater area of negatively charged residues (red, circled) at the N-terminal lobe than in the non-fused system.

both the *Hi*SiaPQM (*Peter et al., 2022*) and *Pp*SiaPQM (*Davies et al., 2023*) systems, highlighting the importance of this area for the transporter. It is clear that this region is important for SiaP recognition (also shown by the adjacent positioning of SiaP to this region in the complex model, *Figure 6—figure supplement 2*) and may be further involved in modulating the conformational changes in SiaQM. Similarly, mutations R30E, S356Y, and E429R all significantly reduced bacterial growth and SiaP binding (*Peter et al., 2022*). These mutations are consistent with disrupting the SiaP:SiaQM interaction as the residues are all solvent-accessible at the periplasmic surface of *Hi*SiaQM in our structure and are close to *Hi*SiaP in the complex model. All three residues are strictly conserved across the TRAP transporter family (*Figure 4—figure supplements 1 and 2*) and our structure shows that R30 (SiaQ) and E429 (SiaM) form a salt bridge (~2.5 Å) between two helices of the scaffold to connect SiaQ and SiaM, highlighting their structural significance (*Figure 6—figure supplement 3*). Overall, the *Hi*SiaPQM complex model is consistent with the mutagenesis of the *Hi*SiaPQM and *Pp*SiaPQM systems (*Peter et al., 2022*; *Davies et al., 2023*), but requires an experimental tripartite complex (SiaPQM) structure for validation.

Finally, the ability of related SiaPQM systems to substitute subunits was assessed using the transport assay (*Figure 5a*; *Figure 7a*). Two fused SiaPQM systems from the Pasteurellaceae family (*H. influenzae* and *Aggregatibacter actinomycetemcomitans*) were able to substitute the QM- and P-subunits and retain activity. With *Hi*SiaQM in the proteoliposome, there was no transport when *Pp*SiaP from the non-fused system was used (pink) but was fully functional when *Aa*SiaP was used (blue). We also substituted the QM-subunits: with *Aa*SiaQM in the proteoliposome, the transport was the same with *Hi*SiaP and *Aa*SiaP but slower in the *Aa*SiaPQM system than *Hi*SiaPQM, suggesting that differences in the mechanisms (e.g., the affinity of subunits or Na$^+$) may exist even in systems similar enough to exchange subunits. That there was no transport with *Hi*SiaQM and *Pp*SiaP follows previous work demonstrating that SiaP from *V. cholerae* did not function with *Hi*SiaQM (*Mulligan et al., 2009*). Structural analysis of the SiaP proteins from the fused and non-fused systems shows subtle differences in surface charge (*Figure 7b*), yet the activity results suggest the interacting surfaces have diverged enough to make them incompatible. It may also be that each system uses slightly different methods to engage the allosteric opening of SiaP.

## Discussion

Using the well-studied *Hi*SiaPQM TRAP as a model system (*Severi et al., 2005*; *Müller et al., 2006*; *Johnston et al., 2008*; *Fischer et al., 2015*; *Peter et al., 2022*; *Mulligan et al., 2009*), we (1) determined a considerably higher resolution structure of the transmembrane subunits of this fused TRAP transporter, allowing us to now resolve side chains, lipids, the Na$^+$ ion binding sites, and the substrate-binding site; (2) demonstrate that *Hi*SiaQM can form a dimeric configuration in maltoside detergents, amphipol and nanodiscs; and (3) determined the nature of the interaction between *Hi*SiaP and *Hi*SiaQM using biophysical methods. These experiments lead to many new conclusions that shape our understanding of how TRAP transporters function.

### The 'elevator down' (inward-facing) conformation is preferred in experimental conditions

The two previous SiaQM structures have relied on the presence of a megabody for structure determination, which made contacts with the periplasmic surface of the transport domain. This interaction may stabilise the protein in the observed 'elevator down' conformation. However, the dimeric structures we have presented have no other proteins bound, yet exist stably in the elevator down state, suggesting this is the most stable conformation in experimental conditions, where there is no membrane bilayer, membrane potential, or chemical gradient present.

### Lipids have strong interactions with *Hi*SiaQM and are likely to be important for the transport mechanism

The structure of the fused *Hi*SiaQM confirms the presence of an extra transmembrane helix in the Q-subunit compared to the non-fused TRAP systems that lack this helix. The extra transmembrane helix connects the small subunit to the large subunit via a short horizontal arm helix that runs parallel to the membrane. This horizontal helix is comprised of several positively charged residues that face

the periplasm, contributing to an extra positive area on the surface of the transporter that is not seen in the non-fused *Pp*SiaQM. The horizontal helix may act similarly to the two arm helices and improve transport by increasing the steadiness of the scaffold of the transporter. The high-resolution structure of *Hi*SiaQM has identified well-defined lipids associated with tryptophan residues of the Q-subunits that form the dimer interface and a newly described single lipid in the binding pocket formed by the fusion helix. Both of these areas are likely important for anchoring the scaffold domain and allowing transport. As the single phospholipid is bound within the protein, it may also have a physiological role in P-subunit-binding dynamics as a mechanism of regulation depending on lipid type.

### *Hi*SiaQM can exist as a monomer or a dimer

We demonstrate that *Hi*SiaQM can adopt both a monomeric and a dimeric configuration depending on the relative concentration of L-MNG detergent, with high concentrations favouring the monomer. This behaviour is characteristic of a self-association (*Figure 2—figure supplement 1a*, *Figure 3a*). The formation of a stable dimeric species is also found in our amphipol structure, other detergents, and nanodiscs.

The parallel dimer is consistent with other elevator transporters, although the interface between the monomers is reduced. Notably, a related TAXI-TRAP transporter also appears to form a self-association (*Roden et al., 2023*) and the dicarboxylate transporter, VcINDY from *V. cholerae,* adopts a stable homodimeric configuration in crystal and solution (*Mulligan et al., 2016*; *Mancusso et al., 2012*). VcINDY uses an elevator mechanism and cross-linking the dimer interface does not perturb transport, implying a lack of coupling between the monomers and an absence of large conformational changes at the interface (*Mulligan et al., 2016*). The dimerisation of VcINDY is used to stabilise the transport domains and allow the elevators to move up and down against the bound interface. Even with a small interface, a dimeric *Hi*SiaQM assembly may further stabilise the scaffold of the two monomeric units.

The antiparallel dimer is hard to rationalise from a physiological standpoint. Although not unprecedented in vivo with small bacterial multidrug transporters (*Lloris-Garcerá et al., 2012*; *Fluman et al., 2017*), an antiparallel structure makes little biological sense in the case of the TRAP transporter. In the antiparallel structure, the inverted monomer displays the cytoplasmic face towards the periplasm and will not bind SiaP, which is required for transport. This structure could result from removing the protein from its native environment, where it is constrained by the cell membrane, and in the absence of the constraints of the membrane can rotate around the interface when solubilised by detergents.

### The affinity of SiaP for SiaQM is weak

For the SiaPQM TRAP transporter, the role of SiaP is to capture Neu5Ac with high affinity and deliver it to SiaQM before leaving to bind more Neu5Ac. As a result, a high-affinity SiaPQM complex that has a slow SiaP off-rate is likely not desired. Neither is a low-affinity SiaPQM complex that has a slow SiaP on-rate, as SiaQM relies on binding SiaP for transport (*Figure 5a*). The $K_D$ observed in our experiments may be representative of a system with a fast off-rate (and potentially a fast on-rate) for SiaP. The resulting $K_D$ is not of high affinity, but the transport cycle is optimised as SiaP quickly binds SiaQM, delivers Neu5Ac, and then leaves without getting 'stuck' to the transporter. The fast off-rate would reflect the conformational coupling of SiaP to SiaQM as the open state of SiaP that occurs after delivering Neu5Ac has been shown to bind SiaQM less (*Peter et al., 2022*).

The determined $K_D$ may also reflect the experimental conditions. If the transporter is often in the conformation we observe to be stabilised in the structures, then the residues on the periplasmic surface of SiaQM may not be accessible or correctly positioned for binding with the partner residues on SiaP. In addition to the effects of detergent, the experimental conditions do not provide a Na$^+$ gradient, which may have significant consequences on the positioning of SiaQM for interaction with SiaP. Under similar conditions, *Peter et al., 2022* estimated a comparable $K_D$ of >1 µM, further indicating that the interaction affinity is low.

### Conclusion

We have determined the oligomeric state of detergent-solubilised *Hi*SiaQM and the structure of amphipol-solubilised *Hi*SiaQM to near atomic resolution. In both membrane mimetics, we have observed a monomer–dimer self-association. The cryo-EM structure reveals the position of the extra

transmembrane helix and shows that it is made of two sections with a positively charged periplasmic surface. This suggests that the extra helix may be involved in the interaction with SiaP, which has a predominantly negatively charged surface where it would likely bind the transporter. Furthermore, this work yields novel insights for drug design, as our structure can guide the design of inhibitors that block the SiaPQM interaction or use the system as a Trojan horse. In bacteria that rely solely on SiaPQM for obtaining Neu5Ac, such as *H. influenzae*, these inhibitors may have utility as antibacterial agents.

## Materials and methods

**Key resources table**

| Reagent type (species) or resource | Designation | Source or reference | Identifiers | Additional information |
|---|---|---|---|---|
| Gene (*Haemophilus influenzae* Rd KW20) | SiaP; SiaQM | UniProt | P44542, P44543 | |
| Strain, strain background (*Escherichia coli*) | BL21 (DE3); TOP10 | Invitrogen | | Chemically competent cells |
| Antibody | Anti-Xpress IgG (mouse monoclonal) | Invitrogen | | WB (1:2500) |
| Antibody | Anti-mouse IgG (rabbit) | Sigma-Aldrich | | WB (1:15000) |
| Recombinant DNA reagent | pET22b(+) (plasmid) | GenScript | | Provides pelB leader sequence |
| Recombinant DNA reagent | pBAD/HisA (plasmid) | Thermo Fisher Scientific | | |
| Peptide, recombinant protein | cNW11 | Now Scientific | | Covalent nanodisc protein |

### Cloning of the substrate-binding protein, SiaP, and the membrane-bound protein, SiaQM

The gene encoding SiaP from *H. influenzae* Rd KW20 was synthesised and cloned into the pET22b(+) vector to include a pelB leader sequence (GenScript). The pelB sequence signals the protein to the periplasm and is cleaved in vivo before purification. The gene encoding SiaQM from *H. influenzae* Rd KW20 was synthesised and cloned into the pBAD/HisA vector to include an N-terminal his-tag, Xpress epitope, enterokinase cleavage site, and a linker sequence (GeneArt, Thermo Fisher Scientific). Full protein sequences for these genes are provided in *Supplementary file 2*.

### Expression and purification of the membrane-embedded protein, SiaQM

*E. coli* TOP10 cells containing the pBAD/HisA_*Hi*SiaQM plasmid were grown in Terrific Broth medium in 1.5 L cultures at 37°C to $OD_{600}$ = 1.4–1.8 before induction with 0.2% L-arabinose. Induced cultures were incubated for 3 hr at 37°C and 180 rpm. Cells were harvested by centrifugation at 8000 × *g* for 10 min at 4°C, snap-frozen, and stored at –80°C. Cells were thawed at room temperature and added to resuspension buffer (1× phosphate-buffered saline [PBS] pH 7.4, 0.5 mg/mL lysozyme and 1 cOmplete protease inhibitor tablet [Roche] per 50 mL solution, or 1 mM phenylmethylsulfonyl fluoride [PMSF]). Cells were resuspended and then lysed by sonication (70% amplitude, 0.5 s on /0.5 s off). Unbroken cells and cell debris were removed by centrifugation at 18,000 × *g* for 25 min at 4°C twice. Membranes were harvested by ultracentrifugation at 210,000 × *g* for 2 hr at 4°C, snap-frozen, and stored at –80°C.

Membranes were resuspended in membrane resuspension buffer (1× PBS pH 7.4, 1 mM PMSF, 5 mM dithiothreitol and 6% v/v glycerol). Once resuspended, 2% w/v L-MNG or DDM was added for solubilisation for 2 hr at 4°C with gentle agitation. Insoluble material was removed by ultracentrifugation at 150,000 × *g* for 1 hr at 4°C. The supernatant was collected, and imidazole was added to a final concentration of 20 mM. The sample was applied to a HisTrap FF 5 mL column (Cytiva) equilibrated with buffer A (70 mM Tris pH 8, 150 mM NaCl, 20 mM imidazole, 6% v/v glycerol, 5 mM β-mercaptoethanol [BME], 0.002–0.020% L-MNG [or 0.0348% DDM], and 0 or 5 mM Neu5Ac). The column was washed with 20 column volumes (CV) buffer A, then 10 CV 95% buffer A

and 5% buffer B (70 mM Tris pH 8, 150 mM NaCl, 500 mM imidazole, 6% v/v glycerol, 5 mM BME, 0.002–0.020% L-MNG [or 0.0348% DDM], and 0 or 5 mM Neu5Ac) before eluting the protein with a gradient to 100% buffer B. Fractions identified by SDS-PAGE to contain *Hi*SiaQM were pooled and concentrated. Concentrated protein was applied to a HiLoad 16/600 Superdex 200 pg size-exclusion column (Cytiva) equilibrated in buffer C (50 mM Tris pH 8, 150 mM NaCl, 0.002–0.020% L-MNG [or 0.0348% DDM], and 0 or 5 mM Neu5Ac). The final purified protein was obtained at a yield of 0.5 mg/L of *E. coli* culture and was at least 95% pure by SDS-PAGE (*Figure 2—figure supplement 1b*).

## Expression and purification of the substrate-binding protein, SiaP

*E. coli* BL21 (DE3) cells containing the pET22b(+) *Hi*SiaP plasmid were grown in M9 minimal medium with 0.1 mg/mL ampicillin in 1 L cultures at 26°C to $OD_{600}$ = 0.3 before induction with 1 mM IPTG. Induced cultures were incubated for 16 hr at 26°C and 180 rpm. Cells were harvested by centrifugation at 8000 × *g* for 10 min at 4°C and resuspended in periplasmic extraction buffer (30 mM Tris pH 8, 20% w/v sucrose and 1 mM ethylenediaminetetraacetic acid [EDTA]) at room temperature (RT). The resuspension was incubated at RT for 10 min with shaking, then centrifuged at 13,000 × *g* for 10 min at 4°C. The supernatant was removed and the cell pellet rapidly resuspended in ice-cold Milli-Q water. The resuspension was incubated for 10 min at 4°C with shaking, then centrifuged as before. The supernatant was collected and PMSF was added to a concentration of 1 mM. The supernatant was filtered with a 0.2 μm filter and applied to a HiPrep Q FF 16/10 column (Cytiva) equilibrated in buffer D (50 mM Tris pH 8). The column was washed with 5 CV of buffer D before eluting the protein with a gradient to 100% buffer E (50 mM Tris pH 8, 1 M NaCl). SDS-PAGE was used to identify fractions containing *Hi*SiaP. These fractions were pooled, and ammonium sulphate was added to a concentration of 1 M. The protein was applied to a HiPrep Phenyl FF 16/10 column (Cytiva) equilibrated in buffer F (50 mM Tris pH 8, 1 M ammonium sulphate). The column was washed with 5 CV of buffer F before eluting the protein with a gradient to 100% buffer G (50 mM Tris pH 8, 150 mM NaCl). SDS-PAGE was used to identify fractions containing *Hi*SiaP. These fractions were pooled, concentrated, and applied to a HiLoad 16/600 Superdex 200 pg size-exclusion column (Cytiva) equilibrated with buffer G. Fractions containing *Hi*SiaP were pooled and concentrated for use in experiments. SDS-PAGE was used to visually assess purity.

## Reconstitution into nanodiscs

The membrane scaffold protein cNW11 (purchased from Now Scientific, Boston, USA) was used for the reconstitution of *Hi*SiaQM into nanodiscs. Lipid stocks of POPC (1-palmitoyl-2-oleoyl-glycero-3-phosphocholine) and POPG (1-palmitoyl-2-oleoyl-sn-glycero-3-phosphoglycerol) in chloroform were aliquoted in a 3:2 ratio and dried under nitrogen in a glass tube. The lipids were dissolved by heating and vortexing with buffer H (20 mM Tris pH 7.4, 100 mM NaCl, 0.5 mM EDTA, and 160 mM sodium cholate). *Hi*SiaQM, cNW11, and lipids were incubated together at the three ratios for 1 hr on ice with a final cholate concentration between 25 and 27 mM. For empty nanodiscs, a similar reconstitution was set up with only cNW11 and lipids. Bio-Beads SM-2 were added (0.5 g to each 500 μL reconstitution) and the solution was incubated on ice for another 30 min before gentle rotation for 16 hr at 4°C. The solution was removed from the Bio-Beads with a 24-gauge needle and spun at 14,000 × *g* to remove insoluble aggregates. The sample was concentrated and applied to a Superdex 200 Increase 10/300 GL column equilibrated with buffer I (20 mM Tris pH 7.4, 100 mM NaCl, and 0.5 mM EDTA). Fractions of interest were concentrated and used for future experiments. SDS-PAGE was used to visually assess the presence of nanodisc-reconstituted *Hi*SiaQM.

## Reconstitution into amphipol

Amphipol A8-35 (Anatrace) was added at 5 mg per 1 mg purified *Hi*SiaQM in L-MNG. Bio-Beads SM-2 were added at 100 mg per mL of the mixture before incubating overnight at 4°C with gentle rotation. Following reconstitution, the amphipol-bound protein was subjected to another size-exclusion chromatography step to remove unincorporated amphipol and any aggregated protein. Protein that eluted at a volume consistent with *Hi*SiaQM was pooled, concentrated, and used in experiments.

## Western blotting

A western blot was used to verify the bands containing *Hi*SiaQM on SDS-PAGE. *Hi*SiaQM was separated by SDS-PAGE and transferred to nitrocellulose using NuPAGE transfer buffer + 20% methanol. The transfer was performed for 15 min at 15 V. Membranes were blocked with Tris-buffered saline + 0.1% Tween 20 (TBST) and 2% bovine serum albumin (BSA) for 1 hr at 4°C. Mouse anti-Xpress IgG monoclonal antibody (Invitrogen) was diluted to 1/2500 in TBST + 2% BSA and incubated with the membranes overnight at 4°C. Membranes were washed with water before incubating with rabbit anti-mouse IgG secondary antibody (Sigma-Aldrich) with conjugated alkaline phosphatase at 1/15,000 in TBST + 2% BSA for 1 hr at RT. Membranes were washed with TBST for 5 min, three times. Detection was performed by incubating the membranes with NBT/BCIP substrate solution (Thermo Fisher Scientific) for 10 min. The reaction was stopped by rinsing the membranes in water.

## Analytical ultracentrifugation

Sedimentation velocity experiments were performed in a Beckman Coulter XL-I AUC instrument unless otherwise stated. Samples were loaded into 12 mm double sector cells with sapphire windows, then mounted in an An-60 Ti rotor. Sedimentation data were analysed with UltraScan 4.0 (*Demeler and Gorbet, 2016*; *Dubbs et al., 2024*). Optimisation was performed by two-dimensional spectrum analysis (2DSA) (*Brookes et al., 2006*; *Brookes et al., 2010*) with simultaneous removal of time- and radially invariant noise contributions and fitting of boundary conditions. Where appropriate, 2DSA solutions were subjected to parsimonious regularisation by genetic algorithm analysis (*Brookes and Demeler, 2007*).

For determining the oligomeric state of *Hi*SiaQM, the samples (0.40 mg/mL in L-MNG, 0.55 mg/mL in DDM) were in buffer G with either 20 times the critical micelle concentration (CMC) of L-MNG (0.02%, 200 μM) or 4 times the CMC of DDM (0.0348%, 0.68 mM). Buffer density and viscosity values were measured with a density meter and microviscometer (Anton Paar). Radial absorbance data were collected at 280 nm, 42,000 rpm and 20°C (L-MNG) or 4°C (DDM). Oligomeric state and bound detergent calculations were performed using a method similar to *Henrickson et al., 2023*, utilising sedimentation and diffusion coefficients obtained from UltraScan (*Demeler and Gorbet, 2016*; *Dubbs et al., 2024*). These coefficients and the partial specific volumes of the detergents (L-MNG = 0.797 mL/g, DDM = 0.82 mL/g) and protein (0.7634 mL/g) were used to calculate the masses of the detergent-bound protein species.

For determining the *Hi*SiaQM species in the nanodisc and amphipol samples, the methods used were similar to that above. The nanodisc samples were in buffer I. The data were collected at 280 nm, 46,000 rpm, and 10°C. The amphipol sample was in buffer G. The data were collected at 0.27 mg/mL, 280 nm (intensity data), 40,000 rpm, and 20°C in a Beckman Coulter Optima AUC instrument at the Canadian Center for Hydrodynamics at the University of Lethbridge.

For determining the oligomeric state of *Hi*SiaP, the samples were in buffer G and buffer G+5 mM Neu5Ac. Buffer density and viscosity values were estimated with UltraScan (*Demeler and Gorbet, 2016*) or measured with a density meter and microviscometer (Anton Paar). The data were collected at 280 nm, 42,000 or 50,000 rpm, and 20°C.

## Fluorescence detection analytical ultracentrifugation

Experiments were performed with FITC-labelled *Hi*SiaP and a concentration series of *Hi*SiaQM. The FITC labelling was optimised to produce a ratio of 0.93 (~1) moles of FITC per mole of protein. A Beckman Coulter XL-A AUC instrument with a fluorescence detection system (AVIV Biomedical) was used with an An-50 Ti rotor at the University of Melbourne, Australia. To generate an artificial bottom, 50 μL of FC43 fluorinert oil was loaded into the bottom of each cell. The samples (350 μL) in buffer G+ 0.002% L-MNG and 5 mM Neu5Ac were loaded into 12 mm double sector cells. *Hi*SiaP was kept constant at 10 nM and *Hi*SiaQM was varied across 14 concentrations (twofold dilutions from 40 μM to 4.9 nM). The samples were incubated for an hour at room temperature, then run at 50,000 rpm and 20°C in fluorescence mode.

## Proteoliposome assays

Purified *Hi*SiaQM was reconstituted using a batch-wise detergent removal procedure as previously described (*Wahlgren et al., 2018*; *North et al., 2018*). In brief, 50 μg of *Hi*SiaQM was mixed with

120 µL of 10% $C_{12}E_8$ and 100 µL of 10% egg yolk phospholipids (w/v) in the form of sonicated liposomes as previously described (*Scalise et al., 2017*), 50 mM of $K^+$-gluconate and 20 mM HEPES/Tris pH 7.0 in a final volume of 700 µL. The reconstitution mixture was incubated with 0.5 g Amberlite XAD-4 resin under rotatory stirring (1200 rev/min) at 25°C for 40 min (*Scalise et al., 2017*).

After reconstitution, 600 µL of proteoliposomes were loaded onto a Sephadex G-75 column (0.7 cm diameter × 15 cm height) pre-equilibrated with 20 mM HEPES/Tris pH 7.0 with 100 mM sucrose to balance the internal osmolarity. Then, valinomycin (0.75 µg/mg phospholipid) prepared in ethanol was added to the eluted proteoliposomes to generate a $K^+$ diffusion potential, as previously described (*Wahlgren et al., 2018*). For a standard measurement, after 10 s of incubation with valinomycin transport was started by adding 5 µM [$^3$H]-Neu5Ac to 100 µL proteoliposomes in the presence of 50 mM $Na^+$-gluconate and 0.5 µM of *Hi*SiaP. Experiments were also performed varying the [$Na^+$-gluconate] and SiaP species to determine their effect on transport. The transport assay was terminated by loading each proteoliposome sample (100 µL) on a Sephadex G-75 column (0.6 cm diameter × 8 cm height) to remove the external radioactivity. Proteoliposomes were eluted with 1 mL 50 mM NaCl and collected in 4 mL of scintillation mixture, vortexed, and counted. The radioactivity taken up in controls performed with empty liposomes, that is, liposomes without incorporated protein, was negligible with respect to the data obtained with proteoliposomes, that is, liposomes with incorporated proteins. All measurements are presented as means ± SEM from independent experiments as specified in the figure legends. Data analysis and graphs of data and fit were produced using GraphPad Prism (version 9).

## Single-particle cryogenic electron microscopy data collection, processing, and map refinement

*Hi*SiaQM was purified as before but with buffers containing a low concentration of L-MNG detergent (0.002%, 20 µM) to produce a population favouring dimeric particles (see *Figure 2—figure supplement 1a*, blue trace). The protein was exchanged into amphipol A8-35, concentrated, and frozen at –80°C until use. The protein was thawed, centrifuged to remove any aggregates, concentrated to 4.1 mg/mL, applied to a glow-discharged (Gatan Solarus) Cu 2/1 carbon Quantifoil grid (Electron-microscopy Sciences), and blotted using a Vitrobot Mark IV (Thermo Fisher Scientific) for 3.5 s, at 4°C, and with 100% humidity before vitrification in liquid ethane. The grid was screened for ice quality and particle distribution on a Talos Arctica microscope. A 14,281-image dataset was collected. The data collection parameters are provided in *Supplementary file 1*.

All cryo-EM data processing was performed using CryoSPARC v.3.2.0 (*Punjani et al., 2017*; *Figure 2—figure supplement 2*). In brief, movie frames were aligned using patch motion correction with a B-factor of 500, and then contrast transfer function (CTF) estimations were made using the patch CTF estimation tool. Initially, 84,871 particles were picked from 295 micrographs using the Blob Picker tool and extracted. These particles were 2D classified into 50 classes, and 10 of these 2D classes were selected and used as a template for automated particle picking using the Template Picker tool, where 3,957,461 particles were picked from 14,281 micrographs. Particles were inspected with the Inspect Picks tool using an NCC Score Threshold of 0.1, and a Local Power range of 29,159–62,161. A total of 2,950,415 particles were extracted with a box size of 400 pixels and Fourier cropped to 200 pixels. The extracted particles were then sorted using iterative rounds of 2D classification, where the best 39 classes showing some structural details were selected, retaining 618,442 particles. The particles were subjected to ab initio reconstruction separated into four classes. The two best 3D reconstructions represent dimers that oligomerised in two different ways—an antiparallel dimer and a parallel dimer, for which we had 225,044 and 220,810 particles, respectively. At this point, the datasets were separated for further processing. For each set of data, the reconstructions were used as a reference model for iterative rounds of non-uniform refinement, allowing a 2.99 Å map to be reconstructed for the antiparallel dimer and a 3.36 Å map to be reconstructed for the parallel dimer. For refinement of the antiparallel dimer, C2 symmetry was imposed. The final rounds of refinement were conducted using unbinned particles. The model was built using an initial model from AlphaFold (*Jumper et al., 2021*), which was fit to the maps using Namdinator (*Kidmose et al., 2019*), ISOLDE (*Croll, 2018*), and Coot (*Emsley et al., 2010*), both used for further model building. A summary of the statistics for data processing, refinement, and validation is shown in *Supplementary file 1*.

## Small-angle X-ray scattering

Small-angle X-ray scattering data were collected on the SAXS/WAXS beamline equipped with a Pilatus 1 M detector (170 mm × 170 mm, effective pixel size, 172 µm × 172 µm) at the Australian Synchrotron. A sample detector distance of 1600 mm was used, providing a $q$ range of 0.05–0.5 Å$^{-1}$. *Hi*SiaP (50 µL at 12.3 mg/mL) was measured in buffer G+0.1% w/v NaN$_3$ and buffer I+0.1% w/v NaN$_3$ and 5 mM Neu5Ac. Each sample was injected onto a Superdex 200 Increase 5/150 GL column (Cytiva) before data collection with 1 s exposures. Buffer subtraction and data analysis were performed in CHROMIXS and PRIMUS from the ATSAS program suite (*Manalastas-Cantos et al., 2021*).

## Software

Molecular graphics and analyses were made with PyMOL (*Schrödinger, LLC, 2024*) and UCSF ChimeraX (*Pettersen et al., 2021*). Figures were made with Adobe Illustrator.

## Acknowledgements

This research was undertaken in part using the MX2 beamlines, and made use of the Australian Cancer Research Foundation (ACRF) detector, as well as the SAXS beamline at the Australian Synchrotron, part of ANSTO. RAN and MJC acknowledge the Canterbury Medical Research Foundation (CMRF) for Major Project Grant funding and the Biomolecular Interaction Centre (UC) for funding support. This research was supported by an AINSE Ltd. Postgraduate Research Award (PGRA) to MJC and JSD, RCJD acknowledges the following for funding support, in part (1) the Marsden Fund, managed by Royal Society Te Apārangi (contract UOC1506); (2) a Ministry of Business, Innovation and Employment Smart Ideas grant (contract UOCX1706); and (3) the Biomolecular Interaction Centre (UC). We thank Yee-Foong Mok and Amy Henrickson for their assistance with AUC experiments. Cryo-EM data were collected at the Cryo-EM Swedish National Facility funded by the Knut and Alice Wallenberg Foundation, the Family Erling Persson and Kempe Foundations, SciLifeLab, Stockholm University, and Umeå University.

## Additional information

### Competing interests

Rosmarie Friemann: Currently employed by AstraZeneca. David Drew: Reviewing editor, *eLife*. The other authors declare that no competing interests exist.

### Funding

| Funder | Grant reference number | Author |
| --- | --- | --- |
| Australian Institute of Nuclear Science and Engineering Ltd. | Postgraduate Research Award | Michael J Currie James S Davies |
| Marsden Fund | UOC1506 | Renwick CJ Dobson |
| Ministry of Business, Innovation and Employment | UOCX1706 | Renwick CJ Dobson |
| Biomolecular Interaction Centre | | Renwick CJ Dobson |

The funders had no role in study design, data collection and interpretation, or the decision to submit the work for publication.

### Author contributions

Michael J Currie, Conceptualization, Data curation, Formal analysis, Supervision, Investigation, Methodology, Writing – original draft, Writing – review and editing; James S Davies, Conceptualization, Data curation, Formal analysis, Supervision, Investigation, Visualization, Methodology, Writing – original draft, Writing – review and editing; Mariafrancesca Scalise, Data curation, Formal analysis,

Investigation, Writing – original draft, Writing – review and editing; Ashutosh Gulati, Software, Validation, Investigation, Methodology; Joshua D Wright, Investigation, Methodology; Michael C Newton-Vesty, Formal analysis, Investigation, Methodology, Writing – review and editing; Gayan S Abeysekera, Formal analysis, Investigation, Methodology; Ramaswamy Subramanian, Soichi Wakatsuki, Supervision, Funding acquisition, Writing – review and editing; Weixiao Y Wahlgren, Investigation, Writing – review and editing; Rosmarie Friemann, Supervision, Funding acquisition; Jane R Allison, Conceptualization, Supervision, Funding acquisition, Validation, Investigation, Writing – review and editing; Peter D Mace, Conceptualization, Resources, Formal analysis, Supervision, Funding acquisition, Validation, Investigation; Michael DW Griffin, Resources, Formal analysis, Supervision, Investigation; Borries Demeler, Conceptualization, Resources, Software, Supervision, Funding acquisition, Validation, Investigation, Methodology, Writing – review and editing; David Drew, Conceptualization, Resources, Formal analysis, Supervision, Funding acquisition, Investigation, Visualization, Writing – review and editing; Cesare Indiveri, Resources, Data curation, Supervision, Validation, Investigation, Methodology; Renwick CJ Dobson, Conceptualization, Resources, Formal analysis, Supervision, Funding acquisition, Investigation, Visualization, Project administration, Writing – review and editing; Rachel A North, Conceptualization, Formal analysis, Supervision, Funding acquisition, Validation, Investigation, Visualization, Project administration, Writing – review and editing

## Author ORCIDs

Michael J Currie ⓘ https://orcid.org/0000-0003-1509-7581
James S Davies ⓘ https://orcid.org/0000-0003-4029-1650
Joshua D Wright ⓘ https://orcid.org/0000-0003-3870-9800
Weixiao Y Wahlgren ⓘ https://orcid.org/0000-0003-0413-165X
Peter D Mace ⓘ https://orcid.org/0000-0003-2175-9537
David Drew ⓘ http://orcid.org/0000-0001-8866-6349
Renwick CJ Dobson ⓘ https://orcid.org/0000-0002-5506-4939

Reviewer #1 (Public Review): https://doi.org/10.7554/eLife.92307.3.sa1
Reviewer #2 (Public Review): https://doi.org/10.7554/eLife.92307.3.sa2
Reviewer #3 (Public Review): https://doi.org/10.7554/eLife.92307.3.sa3
Author Response https://doi.org/10.7554/eLife.92307.3.sa4

## Additional files

### Supplementary files

• Supplementary file 1. Cryo-EM data collection, processing, refinement, and validation statistics.

• Supplementary file 2. Protein sequences of *Hi*SiaQM and *Hi*SiaP expressed and purified in this work.

• MDAR checklist

### Data availability

Structural coordinates have been deposited in PDB under accession codes 8THI and 8THJ. Cryo-EM data had been deposited to EMDB under accession codes EMD-41265 and EMD-41266. All other data in this study are provided within the manuscript.

The following datasets were generated:

| Author(s) | Year | Dataset title | Dataset URL | Database and Identifier |
|---|---|---|---|---|
| Davies JS, Currie MC, Dobson RCJ, North RA | 2023 | Cryo-EM structure of the Tripartite ATP-independent Periplasmic (TRAP) transporter SiaQM from Haemophilus influenzae (parallel dimer) | https://www.rcsb.org/structure/8THI | RCSB Protein Data Bank, 8THI |
| Davies JS, Currie MC, Dobson RCJ, North RA | 2023 | Cryo-EM structure of the Tripartite ATP-independent Periplasmic (TRAP) transporter SiaQM from Haemophilus influenzae (antiparallel dimer) | https://www.rcsb.org/structure/8THJ | RCSB Protein Data Bank, 8THJ |
| Davies JS, Currie MC, Dobson RCJ, North RA | 2023 | Cryo-EM structure of the Tripartite ATP-independent Periplasmic (TRAP) transporter SiaQM from Haemophilus influenzae (parallel dimer) | https://www.ebi.ac.uk/emdb/EMD-41265 | EMDataBank, EMD-41265 |
| Davies JS, Currie MC, Dobson RCJ, North RA | 2023 | Cryo-EM structure of the Tripartite ATP-independent Periplasmic (TRAP) transporter SiaQM from Haemophilus influenzae (antiparallel dimer) | https://www.ebi.ac.uk/emdb/EMD-41266 | EMDataBank, EMD-41266 |

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
