## [Editor Report · eLife assessment]

This article presents a cryo-EM structure of a tripartite ATP-independent periplasmic (TRAP) transporter that contributes to *Haemophilus influenzae* virulence. **Convincing** biophysical and cryo-EM experiments yield a **valuable** molecular model, but the functional importance of some of the molecular features identified remains to be demonstrated.

---

## [Referee Report · Reviewer #1 (Public Review)]

Summary:

TRAP transporters are an unusual class of secondary active transporters that utilize periplasmic binding proteins to deliver their substrates. This paper contributes a new 3 Å structure of the Haemophilus influenzae TRAP transporter. The structure joins two other recent cryo-EM structures of TRAP transporters, including a lower resolution structure of the same H. influenzae protein (overall 4.7 Å), and a ~3 Å structure of a homologue from P. profundum. In addition to reporting a higher resolution cryo-EM structure, the authors also recapitulate protein activity in a reconstituted system, investigate protein oligomerization using analytic ultracentrifugation, and evaluate interactions and function in "mix and match" configurations with periplasmic subunits from other homologues.

Strengths:

The strength of the paper is that the better resolution cryo-EM data permits sidechain assignment, the identification of bound lipids, and the identification of sodium ions. It is important to get this structure out there, since the resolution passes an important threshold for model building accuracy. The current structure nicely explains a lot of prior mutagenesis data on the H. influenzae TRAP. This is also the first structure of a TRAP protein to be solved without a fiducial, although the overall structure is not very different than those solved with fiducials.

Weaknesses:

The experiments examining the monomer/dimer equilibrium appear somewhat preliminary. The biological or mechanistic importance of oligomerization is not established, so these experiments are inherently of limited scope. Moreover, cryo-EM datasets exhibit both parallel and antiparallel dimers, the latter of which are clearly not biologically relevant. It is probably impossible to distinguish these in the AUC experiments, which makes interpretation of these experiments more difficult.

Similarly, the importance of the lipid binding sites observed in cryo-EM aren't experimentally established (for example by mutating the binding site) and it is thus unknown whether they are important for function (as the authors acknowledge).

---

## [Referee Report · Reviewer #2 (Public Review)]

Summary:

In this manuscript, the membrane component of the sialic acid-specific TRAP transporter, SiaQM (HiSiaQM), from H. influenzae, is structurally characterized. TRAP transporters are substrate binding protein (SBP)-dependent secondary-active transporters, and HiSiaQM is the most comprehensively studied member of this family. While all previous work on fused TRAP transporter membrane proteins suggests that they are monomeric (including the previous structural characterization of HiSiaQM by a different group), a surprising finding from this work is the observation that HiSiaQM can form higher oligomers, consistent with it being a dimer. These higher oligomeric states were initially observed after extraction of the protein with LMNG detergent, but were also observed in DDM detergent, amphipol and nanodiscs using analytical ultracentrifugation (AUC). Structural characterization of dimeric HiSiaQM revealed 2 arrangements, a parallel and antiparallel arrangements, the latter of which is unlikely to be physiologically relevant.

The higher resolution of this new structure of HiSiaQM (2.2-2.7 Å compared to 4.7 Å for the previous structure) facilitated the assignment of bound lipids at the dimer interface and a lipid molecule embedded in each of the protomers; allowed for a clearer refinement of the Na+ and putative substrate binding sites, which differ slightly from the previous structure; and produced better modelled side chains for the residues involved in the SBP:HiSiaQM interaction. The authors developed a useful AUC-based assay to determine the affinity for this interaction revealing an affinity of 65 µM. Finally, the authors make the very interesting observation that a sialic acid specific SBP from a different TRAP transporter can utilize HiSiaQM for transport, contrary to previous observations, revealing for the first time that TRAP membrane components can recognize multiple SBPs.

Overall, this is a well written and presented manuscript detailing some interesting new observations about this interesting protein family. One of the main findings, that the protein can form a dimer, is supported by data, but the physiological relevance of this is questionable, and the possibility that this is artefactual has not been ruled out. Conclusions regarding the mechanistic importance of the lipid bindings sites is not currently supported by the data.

Strengths:

The main strength of this work is the increased resolution of HiSiaQM, which allows for much more precise assignment of side chains and their orientation. This will be of importance for subsequent mechanistic studies on the contributions of these residues to Na+ and sialic acid binding and conformational changes.

The observation of the lipids, especially the lipid embedded near the fusion helix, is an intriguing observation, which lays the groundwork for future work to understand the lipid-dependence of these transporters.

The development of the AUC-based approach to measure SBP affinity for the membrane component will likely prove be useful to future studies.

Weaknesses:

One of the main results from this work is the observation that HiSiaQM can form a dimer. Two arrangements were observed, parallel and antiparallel, the latter of which is almost certainly physiologically irrelevant as it would preclude essential interactions with the extracytoplasmic substrate binding protein. As acknowledged by the author, this non-physiological arrangement is likely a consequence of protein preparation (overexpression, extraction, purification, etc.). However, if one dismisses the antiparallel arrangement as non-relevant and an artefact of protein preparation, it is difficult for the parallel arrangement to maintain its credibility, as it was also processed in the same way. This is especially true when one considers that there is only 100 Å2 buried surface area in the parallel arrangement that does not involve any sidechains; it is difficult to envisage this as a specific interaction, e.g. compared to related proteins that have ~2000 Å2 buried surface area. Unless this dimerization is observed in a bacterial membrane at physiological protein concentrations, it is difficult to rule out the possibility that the observed dimerization is merely an artefact caused by the expression, purification and concentration of the protein.

The manuscript contains some excellent structural analysis of this protein, whose higher resolution reveals some new and interesting insight. However, a weakness of the current work is a lack of validation of these observations using other approaches. For example, lipid interactions are observed in the structure that the authors claim is mechanistically important. However, without disrupting these interactions to look at the effect on transport, this conclusion is not supported. Similarly, the authors use their structure to predict residues that are important for the SBP:membrane protein interaction, and they develop an AUC-based binding assay to study this interaction, but they do not test their predictions using this approach.

---

## [Referee Report · Reviewer #3 (Public Review)]

Summary:

The manuscript reports new molecular characterization of the Haemophilus influenza tripartite ATP-independent periplasmic (TRAP) transporter of N-acetylneuraminate (Neu5Ac). This membrane transporter is important for the virulence of the pathogen. H. influenza lacks Neu5Ac biosynthetic pathway, and utilizes the TRAP transporter to import it. Neu5Ac is used as a nutrient source but also as a protection from human immune response. The transporter is composed of two fused membrane subunits, HiSiaQM, and one soluble, periplasmic subunit HiSiaP. HiSiaP, by binding to the substrate Neu5Ac, changes its conformation, allowing its binding to HiSiaQM, followed by Neu5Ac and Na+ transport to the cytoplasm. The combination of structural, biophysical and biochemical approaches provides a solid basis for describing the functioning of the Haemophilus influenza Neu5Ac TRAP transporter, which is essential for the pathogen virulence.

Strengths:

The paper describes the electron microscopy structure of HiSiaQM, thanks to its solubilization in L-MNG followed by exchange to amphipol or nanodisc. In these conditions, HiSiaQM consists in a mixture of monomers and dimers, as characterized by analytical ultracentrifugation. The cryo-EM analysis shows two types of dimers: one in an antiparallel configuration, which is artifactual, and a parallel one, which may be physiologically relevant. Cryo-EM on the dimers allows high resolution (≈ 3 Å) structure determination. The structure is the first one of a fused SiaQM, and is the first obtained without megabody. The work highlights structural elements (fusion helix, lipids) that could modulate transport. The authors checked the functionality of the purified HiSiaQM, which, after reconstitution in liposome, displays a significantly larger Neu5Ac transport activity compared to the non-fused PpSiaQM homolog. The work identifies Na+ binding sites, and the putative Neu5Ac binding site. From analytical ultracentrifugation using fluorescently labelled HiSiaP, the authors show that HiSiaP is able to interact with HiSiaQM monomer and dimer, with a low but physiologically relevant affinity. HiSiaP interaction with HiSiaQM was modelled using AlphaFold2, and discussed in view of published activity on mutants, and new transport activity assays using SiaQM and SiaP from different organisms. In conclusion, the combination of structural, biophysical and biochemical approaches provides a solid basis for describing the functioning of this TRAP fused transporter.

Weakness: This work evidences in vitro a HiSiaQM dimer, whose in vivo relevance is not ascertained. However, the authors are very careful, they do not to over-interpret their data, and their conclusions regarding the transporter structure and function are valid irrespective of its state of association.

---

## [Author Response]

The following is the authors’ response to the original reviews.

**Reviewer #1 (Recommendations For The Authors):**
Congratulations on the very nice structure! In my opinion, which you can feel free to take or leave, this would work better as a short report focused on the improvement of the structure relative to the current published model. To my mind, while the functional and dimerization studies are supportive of the cryo-EM studies (specifically, the purified protein is functional, and does tend to dimerize in various membrane mimetics), these experiments don't provide a lot of new mechanistic insight on their own. The dimerization, in particular, could be developed further.

Response: Thank you for the comments. We have chosen to stick with the current article format. That the protein is dimeric is exciting in our view and we are working to further define the functional significance of this formation.

**Reviewer #2 (Recommendations For The Authors):**
Ln 48. Abstract. "highlighting feature of the complex interface" sounds a bit vague. I was wondering if the authors considered including more specific findings here.

Response: This sentence has been removed.

Ln 149 and elsewhere. The authors refer to the previously published structure of HiSiaQM as "low resolution". It may just be me and likely not the intention of the authors, but this comes across as an attempt to diminish the validity of this previous work from another group, which is not necessary. I would recommend rewording these parts slightly, even if it is just to say "lower resolution" instead of "low resolution".

Response: It was not our intention to diminish the excellent work published by another group, we have changed “low resolution” to “lower resolution” throughout.

Ln 160. The authors state that the inward-open conformation is likely "the resting state of the transporter". I think this statement should be modified slightly to acknowledge that this is only true under these conditions, i.e. in the absence of the bilayer, membrane potential and chemical gradients.

Response: We have edited this as follows “That we observe the inward-open conformation without either a bound P-subunit or fiducial marker, suggests that this is the resting state of the transporter under experimental conditions (in the absence of a membrane bilayer, membrane potential and chemical gradients).”

Ln 202. I'm not convinced that the use of the word "probable" is appropriate here; "possible" would likely fit better in the absence of compelling evidence that this dimer forms in a bacterial cell membrane with physiological levels of HiSiaQM expression.

Response: We have changed “probable” to “possible”.

The authors show an SEC trace for DDM solubilised protein, which is a single peak, whereas the LMNG extracted protein has 2 distinctly different elution profiles depending on the LMNG concentration. Was the same phenomenon observed when varying the DDM concentration?

Response: We observed significantly more aggregation with DDM than L-MNG, so it was infrequently used and considerably less well characterised. In one purification, moderately higher DDM shifted the elution peak to be slightly later but retained a similar profile. Overall, we did not observe the same phenomenon of distinctly different elution profiles with DDM, but we have limited data.

Ln 245. The two positions cited as important for the elevator-type mechanism are the fusion helix and the dimer interface. However, there is no evidence that the dimer interface observed in this work has any relevance to the transport mechanism. To make this statement, the interface would need to be disrupted and the effects on transport evaluated.

Response: This has been edited as follows. “Evident in our cryo-EM maps are well-defined phospholipid densities associated with areas of HiSiaQM that may be important for the function of an elevator-type mechanism (Figure 4), but require further testing.”

Ln 257. The authors state that the lipids form "specific and strong interactions" with the protein, but without knowing the identity of the lipids present, it is difficult to say anything about the specificity of this interaction. I think the authors could consider rewording this.Response: We have edited this by removing the term “specific” and describing the lipidinteractions only as strong interactions.Ln 270. The authors identify a lipid-binding site and residues that likely interact with the headgroup. It would be interesting if the authors could speculate on the purpose of this lipid binding site and how it could affect transport. The residues are not conserved, which the authors suggest reflects the variety of lipid compositions in different bacteria. Are the authors suggesting that this lipid binding site is a general feature for all fused TRAP transporters and that the identity of the lipid changes depending on the species?

Response: Yes, we speculate that the lipid binding site may be a general feature for fused TRAP transporters. We have added speculation about this binding site, specifically that “the fusion helix and concomitant lipid molecule may provide a more structurally rigid scaffold than a Q-M heterodimer, i.e., PpSiaQM, although how this impacts the elevator transition requires further testing” at Line 283.

Though we believe that a binding pocket is likely found in a number of fused TRAPs (based on sequence and Alphafold predictions, e.g., FnSiaQM and AaSiaQM), we have now acknowledged that some fusions may not necessarily bind a lipid molecule here, by stating “While this binding pocket is likely found in a number of fused TRAPs (based on sequence predictions, e.g., FnSiaQM and AaSiaQM in Supplementary Figure 8), it is not clear whether they also bind lipids here without experimental data” at Line 290.

Ln 306. The authors state that the HiSiaPQM has a 10-fold higher transport activity than PpSiaPQM. Unless the transport assays were performed in parallel (to mitigate small changes in experimental set-up) and the reconstitution efficiency for each proteoliposome preparation was carefully analysed, it is very difficult for this to be a meaningful comparison. Even if the amount of protein incorporated into the proteoliposomes is quantified (e.g. by evaluating protein band intensity when the proteoliposomes are analysed using SDS-PAGE), this does not account for an inactive protein that was incorporated, nor the proportion of the protein that was incorporated in the inside-out orientation, which would be functionally silent in these assays. I'm not suggesting these assays actually need to be performed, but I think the text should be modified to reflect what can actually be compared.

Response: We agree with the reviewer that a meaningful comparison is difficult to make without a careful analysis of the reconstitution efficiency and have modified the text to reflect this. We have altered the paragraph beginning at Line 319 to the following: “The fused HiSiaPQM system appears to have a higher transport activity than the non-fused PpSiaPQM system. With the same experimental setup used for PpSiaPQM (5 μM Neu5Ac, 50 μM SiaP) (33), the accumulation of [3H]-Neu5Ac by the fused HiSiaPQM is ~10-fold greater. Although this difference may reflect the reconstitution efficiency of each proteoliposome preparation, it is possible that it has evolved as a result of the origins of each transporter system—P. profundum is a deep-sea bacterium and as such the transporter is required to be functional at low temperatures and high pressures… ”

Ln 335. "S298A did not show an effect on growth when mutated to alanine previously." Suggest changing "S298A" here to "S298".

Response: This has been changed.

Ln 340. In addition to PpSiaQM, the large cavity was also presumably observed in the lower resolution structure of HiSiaQM?

Response: The cavity is detectable in the lower resolution structure (7qe5), though very poorly defined by the density. Furthermore, the AlphaFold model fitted to this density has positioned sidechains inside the cavity, which we consider very likely to be an error (in comparison to our structures, VcINDY and our estimates of the volume required to house sialic acid). The cavity is generally much better defined by the structures we have referenced.

Ln 345. Reference missing after "previously reported"? Response: This has been added.Measuring the affinity for the P-to-QM interaction is very useful, but it would have enhanced the study if some of the residues identified as important for this interaction (detailed on p.13) had been tested for their contributions to binding using this approach.

Response: We do aim to perform this assay with these mutants in the future, but are also developing parallel assays to further test this interaction in different membrane mimetics.

Ln 436. As stated previously, it is more accurate to say that "this is the most stable conformation" under these conditions.

Response: We have edited this to say “The ‘elevator down’ (inward-facing) conformation is preferred in experimental conditions”. We have also changed the last sentence of this paragraph to say “However, the dimeric structures we have presented have no other proteins bound, yet exist stably in the elevator down state, suggesting this is the most stable conformation in experimental conditions, where there is no membrane bilayer, membrane potential, or chemical gradient present.”

Ln 438. "Lipids associated with HiSiaQM are structurally and mechanistically important." This conclusion is not supported by the data presented; there is no evidence that the bound lipids influence the mechanism at all. The lipids observed are certainly interestingly placed and one could speculate about their relevance, but this statement of fact is not supported. Therefore, their importance to the mechanism needs to be tested or this conclusion needs to be substantially softened.

Response: We have softened this statement by changing it to “Lipids have strong interactions with HiSiaQM and are likely to be important for the transport mechanism.”

**Reviewer #3 (Recommendations For The Authors):**
The fact that HiSiaQM samples consist of a mixture of compact monomer and dimer is clear, from Fig. S5 and S6. However, the analysis displayed in Fig 3 and Fig S4 would require more explanation. To my understanding, it requires the values of the sedimentation and diffusion coefficients. It could be good to provide the experimental values of D, and explain a little more about the method in the material and method section.

Response: Yes, the analysis requires the experimental diffusion coefficients. These have been added to the Figure 3 and S4 legends and more detail has been added to the method section.

In addition, I am puzzled when reading, in the legend of Fig 3, considerations that peak 2 could not correspond to a monomer or trimer: do these sentences correspond to other mathematical solutions, or is a given frictional ratio considered, or do they refer to Fig. S5 analysis?

We can see where this confusion could arise from. These sentences do not correspond to a given frictional ratio or the Fig. S5 analysis (this is a separate, complementary analysis). For peak 2 not existing as a monomer is strictly a physical justification – with pure protein and an observed peak smaller than peak 2, a monomer is not possible for peak 2. For peak 2 not existing as a trimer is a mathematical solution using the s and D coefficients. The solutions identify that an unreasonably low amount of detergent would be bound to a trimer (32 molecules for L-MNG or 0 for DDM) to exist at those s and D values so we have ruled the trimer out. Reassuringly, the complementary analysis in Fig. S5/S6 agrees with the monomer-dimer outputs from the s and D analysis. We have adjusted the text in the legends of Fig. 3 and S4 to better convey these points.